JOURNAL OF
Neuroscience Research

# Connectome analysis of male world-class gymnasts using probabilistic multishell, multitissue constrained spherical deconvolution tracking

Hiroyuki Tomita[1]  |  Koji Kamagata[2]  |  Christina Andica[2] (iD)  |  Wataru Uchida[2]  |
Makoto Fukuo[1]  |  Hidefumi Waki[1]  |  Hidenori Sugano[3]  |  Yuichi Tange[3]  |
Takumi Mitsuhashi[3]  |  Matthew Lukies[4]  |  Akifumi Hagiwara[2]  |  Shohei Fujita[2]  |
Akihiko Wada[2]  |  Toshiaki Akashi[2]  |  Syo Murata[2]  |  Mutsumi Harada[1]  |  Shigeki Aoki[2]  |
Hisashi Naito[1]

[1]Juntendo University Graduate School of Health and Sports Science, Chiba, Japan

[2]Department of Radiology, Juntendo University Graduate School of Medicine, Tokyo, Japan

[3]Department of Neurosurgery, Juntendo University Graduate School of Medicine, Tokyo, Japan

[4]Department of Diagnostic and Interventional Radiology, Alfred Health, Melbourne, VIC, Australia

**Correspondence**
Koji Kamagata, Department of Radiology, Juntendo University School of Medicine, 2-1-1 Hongo, Bunkyo-ku, Tokyo 113-8421, Japan
Email: kkamagat@juntendo.ac.jp

**Funding information**
This study was partially supported by the Private University Research Branding Project (Ministry of Education, Culture, Sports, Science and Technology, Japan) and JSPS KAKENHI (grant JP18K09005, JP18H02772)

## Abstract

In athletes, long-term intensive training has been shown to increase unparalleled athletic ability and might induce brain plasticity. We evaluated the structural connectome of world-class gymnasts (WCGs), as mapped by diffusion-weighted magnetic resonance imaging probabilistic tractography and a multishell, multitissue constrained spherical deconvolution method to increase the precision of tractography at the tissue interfaces. The connectome was mapped in 10 Japanese male WCGs and in 10 age-matched male controls. Network-based statistic identified subnetworks with increased connectivity density in WCGs, involving the sensorimotor, default mode, attentional, visual, and limbic areas. It also revealed a significant association between the structural connectivity of some brain structures with functions closely related to the gymnastic skills and the D-score, which is used as an index of the gymnasts' specific physical abilities for each apparatus. Furthermore, graph theory analysis demonstrated the characteristics of brain anatomical topology in the WCGs. They displayed significantly increased global connectivity strength with decreased characteristic path length at the global level and higher nodal strength and degree in the sensorimotor, default mode, attention, and limbic/subcortical areas at the local level as compared with controls. Together, these findings extend the current understanding of neural mechanisms that distinguish WCGs from controls and suggest brain anatomical network plasticity in WCGs resulting from long-term intensive training. Future studies should assess the contribution of genetic or early-life environmental factors in the brain network organization of WCGs. Furthermore, the indices of brain

---

Hiroyuki Tomita and Koji Kamagata contributed equally to the manuscript. Edited by Sandra Chanraud, David McArthur, and Junie Warrington. Reviewed by Ryan Cabeen.

topology (i.e., connection density and graph theory indices) could become markers for the objective evaluation of gymnastic performance.

**KEYWORDS**

brain plasticity, diffusion MRI, graph theory, motor skills, network-based statistic, probabilistic tractography

## 1 | INTRODUCTION

Artistic gymnastics has been included in the Summer Olympics since the first modern Olympic Games in 1896 and has attracted public attention. The unparalleled athletic ability of world-class gymnasts (WCGs) attracts an audience and has raised the following question: "What is the difference between the brains of athletes and novices?" In recent years, elite performance has been considered to be a skill that arises from the neuroplasticity achieved through long-term intensive exercise training (Yarrow et al., 2009). WCGs undergo intensive training from an early age and acquire tremendous motor skills. Genetic predispositions can make individual differences in athletic performance learning (Ahmetov et al., 2016; Guth & Roth, 2013; Yan et al., 2016), but there is no doubt that achieving incredible athletic performance requires long and intensive training. It has been suggested that prolonged intensive exercise training may cause neuronal plasticity in the brain (Dayan & Cohen, 2011; Nakata et al., 2010). Based on this neuroplasticity, WCGs may gain extraordinary abilities in perception, stimulus discrimination, decision-making, exercise preparation, and motor performance. In fact, we have reported that the volume of brain gray matter (GM) in areas related to spatial perception, vision, working memory, and motor control in WCGs is significantly higher than in nonathletes (Fukuo et al., 2020).

A promising technique referred to as *connectome* has been introduced as a method for evaluating brain plasticity in WCGs (Huang et al., 2018; Wang et al., 2013). The connectome is a network representation of whole-brain connectivity that can be mapped to reveal circuit-based alterations in the brain (Fornito et al., 2015). For example, the authors of a recent study used data from diffusion-weighted magnetic resonance imaging (DW-MRI) to infer structural brain connectivity to evaluate the connectome in WCGs (Wang et al., 2013), and they discovered an increase in the density of the structural connectivity between brain regions related to sensorimotor, attentional, and default modes in the WCGs as compared with controls. Graph theory analyses have also identified increased global and local efficiency as well as decreased characteristic path length in the brain network of the WCGs as compared with controls (Wang et al., 2013). Moreover, local network measures have been shown to be changed in WCGs, including increased "functional segregation," as indicated by increased nodal degree, and increased "functional importance," as indicated by increased regional efficiency in the brain area related to motor and attention functions. However, the relationship between each of the

**Significance**

Recent evidence showed that long-term gymnastic training might induce brain neural plasticity, resulting in enhanced performance and higher skill levels in world-class gymnasts (WCGs). To extend the current understanding of neural mechanisms that distinguish WCGs from nonathletes, we evaluated brain neural network "connectome" in 10 WCGs and 10 nonathletes. Our results suggest the plasticity of brain network topology in WCGs resulting from long-term intensive training. Our findings also indicated the association between increased structural connectivity in some brain structures and specific gymnastic skills as indicated by D-score for each apparatus (i.e., floor exercise, pommel horse, vaulting horse, and parallel bars).

six events within men's gymnastics (floor exercise, pommel horse, rings, vault, parallel bars, and horizontal bar) and the structural connectome is unknown. Furthermore, the study mentioned above used deterministic tractography to reconstruct the trajectories of white matter (WM) streamlines and map structural connectomes (Wang et al., 2013). A major limitation of deterministic tractography based on diffusion tensor imaging is the difficulty in accurately estimating neural fiber connections in a voxel, including crossing or kissing fibers (Basser et al., 2000). In contrast, a probabilistic tractography algorithm was proposed to overcome this limitation by estimating multiple fiber directions. Furthermore, tractography is known to be heavily affected by the quality of the diffusion MRI acquisition, leading to false-positive or false-negative connections (Schilling et al., 2019). The use of probabilistic tractography with a new method called multishell, multitissue constrained spherical deconvolution (MSMT-CSD) (Jeurissen et al., 2014), which is based on multishell diffusion MRI, has been shown to increase the precision of tractography at the GM–WM interface as compared with to single-shell, single tissue CSD.

In this study, we aimed to clarify the difference between the brain network structure of gymnasts from that of nonathletes using connectome analysis with MSMT-CSD, which is a novel method of probabilistic tractography. We also aimed to identify the association between changes in brain structural connectivity and the specific abilities of WCGs in different gymnastics events, as marked by the score generated during the championship event.

 

## 2 | PARTICIPANTS AND METHODS

### 2.1 | Participants

The current study was approved by the Ethics Committee of Juntendo University, and written informed consent was acquired from all participants before examination.

This study included 10 Japanese WCGs (all men; mean age, 19.9 ± 1.3 years; age range, 18–22 years; mean years of training, 13.6 ± 2.2 years; years of training range, 10–19 years) who have won at least one medal at Gymnastics World championships. For a comparison, we also included 10 gender- and age-matched healthy controls without any history of gymnastics training or competition (all men; mean age, 20.6 ± 1.7 years; age range, 16–22 years). All subjects were right-handed and had no history of neurologic or psychiatric disease. Note that these subjects are the same subjects targeted in a prior study (Fukuo et al., 2020). However, our prior study evaluated the difference in brain GM volume between gymnasts and nonathletes, and its purpose and method were completely different from that of the present study, which evaluates the difference in brain connectome.

During the competition, gymnastics performances are evaluated with a D-score for difficulty and E-score for execution. The D-score is based on the difficulty value, composition requirements, and connection value of the performance, and the E-score is defined by evaluating technical errors in the routine. The total score is then calculated as the sum of the D-score and the E-score (FIG, http://www.gymnastics.sport/site/). The E-score is influenced by the referees' subjectivity and is more susceptible to the gymnast's mental and physical conditions during competition. Thus, the D-score is a more objective index of the gymnast's specific physical abilities in each event acquired through long-term training. Table 1 lists the D-scores of the Japanese WCGs in each gymnastics event (floor exercise, pommel horse, rings, vault, parallel bars, and horizontal bar) at the most recent world gymnastics competition.

### 2.2 | Image acquisition

Structural MRI and DW-MRI were performed using a 3T MR scanner (MAGNETOM Prisma; Siemens Healthcare, Erlangen, Germany) with a 64-channel head coil. Multishell DW-MRI was acquired using a simultaneous multislice echo-planar imaging sequence in the anteroposterior phase-encoding direction with the following parameters: $b$-values = 1,000 and 2,000 s/mm$^2$ complemented with a non-DW volume ($b = 0$ s/mm$^2$), 64 gradient directions, repetition time (TR) = 3,300 ms, echo time (TE) = 70 ms, voxel size =1.8 × 1.8 × 1.8 mm$^3$, number of slices = 65, simultaneous multislice factor = 2, number of excitations = 1, and acquisition time = 6.25 min. To correct for echo-planar imaging distortions, standard and reverse phase-encoded blipped non-DW-MRI was also obtained. We visually reviewed DW-MRI data in all three orthogonal views to ensure that they were artifact

free, such as missing signals, gross geometric distortion, or bulk motion. Finally, the DW-MRI data were corrected for eddy currents, susceptibility-induced geometric distortions, and intervolume motion using the EDDY and TOPUP toolboxes (Andersson et al., 2016).

To enable the estimation of intracranial volume (ICV) and cortical parcels and tissue segmentation by FreeSurfer, T1-weighted images (T1WIs) were also acquired using three-dimensional (3D) magnetization-prepared rapid gradient echo sequence. The acquisition parameters were as follows: TR = 15 ms, TE = 3.54 ms, inversion time = 1,100 ms, voxel size = 0.86 × 0.86 × 0.86 mm$^3$, and acquisition time = 5.14 min.

### 2.3 | Preprocessing for connectome

We performed the following connectome analyses using the same method as in our previous study (Kamagata et al., 2018). The preprocessing method outlined in Figure 1 was followed, using the Functional MRI of the Brain (FMRIB) Software Library, version 5.0.9 (Greve & Fischl, 2009). The 3D-T1WI was first processed using boundary-based registration to align the image for each subject to the relevant b0 map. Next, non-brain tissue was deleted from each 3D-T1WI using the Brain Extraction Tool (Smith, 2002). Then, the partial volume fractions of WM, cortical GM, and cerebrospinal fluid (CSF) were calculated using the FMRIB Automated Segmentation Tool (Zhang et al., 2001). Further, the partial volume fractions of deep GM in the brain were calculated for all voxels using the FMRIB Integrated Registration and Segmentation Tool (Patenaude et al., 2011). Lastly, each WM, cortical GM, deep GM, and CSF partial volume fraction map was processed for the MSMT-CSD and anatomically constrained in the tractography (ACT) framework. The "5tt2gmwmi" command in the MRtrix3 software package (https://www.mrtrix.org) (Tournier et al., 2019) was utilized to acquire the GM–WM interface mask. While WM masks are often used for seeding, uniformly seeding streamlines through all WM may cause overreconstruction of the streamline density for longer fiber pathways (Yeh et al., 2016). Therefore, GM–WM interface was used as a seeding point for tractogram generation in this study, which is a reported solution to the overreconstruction problem (Girard et al., 2014) but can cause underestimation of the prevalence of long-distance fibers (Zalesky & Fornito, 2009).

### 2.4 | Defining nodes

A total of 84 brain nodes were acquired utilizing the default FreeSurfer pipeline (Dale et al., 1999), based on the Desikan-Killiany cortical atlas segmentation (Desikan et al., 2006). Subsequently, subcortical GM segmentations were obtained from subcortical GM partial volume maps using the FMRIB Software Library Integrated Registration and Segmentation Tool (as outlined previously in the "Preprocessing" section), as there is a high variability in spatial

**TABLE 1** Demographic characteristics of world-class gymnasts

| World-class gymnasts | Best medal record | Age (years) | Years of training | BMI | ICV (ml) | D-score | | | | | | |
|---|---|---|---|---|---|---|---|---|---|---|---|---|
| | | | | | | Floor exercise | Pommel horse | Rings | Vault | Parallel bars | Horizontal bar | Mean ± SD |
| 1 | DTB team challenge 2017 bronze medal. | 20 | 14 | 20.6 | 1,457.6 | 5.3 | 6.0 | 5.6 | 5.2 | 5.8 | 5.6 | 5.58 ± 0.30 |
| 2 | WC 2018 Tokyo cup silver medal (WGC 2018 bronze medal) | 21 | 14 | 21.0 | 1,459.8 | 6.0 | 5.8 | 6.0 | 5.6 | 6.2 | 5.1 | 5.78 ± 0.39 |
| 3 | WGC 2015 gold medal (WGC 2018 bronze medal) | 21 | 12 | 22.5 | 1,623.3 | 5.9 | 6.4 | 6.1 | 5.2 | 6.3 | 5.7 | 5.93 ± 0.44 |
| 4 | Asian Games 2018 silver medal Universiade 2017 gold medal | 21 | 19 | 22.5 | 1,520.3 | 6.0 | 6.0 | 5.7 | 5.2 | 6.2 | 5.7 | 5.80 ± 0.35 |
| 5 | WGC 2015 gold medal | 22 | 13 | 22.04 | 1,582.4 | 6.2 | 5.4 | 5.3 | 5.2 | 5.4 | 5.9 | 5.57 ± 0.39 |
| 6 | DTB team challenge 2018 bronze medal | 20 | 10 | 20.8 | 1,502.7 | 5.9 | 6.3 | 5.6 | 5.2 | 5.7 | 5.2 | 5.65 ± 0.42 |
| 7 | DTB team challenge 2018 bronze medal | 19 | 14 | 24.0 | 1,499.2 | 5.5 | 5.1 | 5.6 | 5.2 | 5.9 | 6.1 | 5.57 ± 0.39 |
| 8 | Voronin cup 2016 gold medal | 19 | 14 | 23.4 | 1,349.7 | 5.6 | 5.8 | 5.5 | 4.8 | 6.0 | 5.8 | 5.58 ± 0.42 |
| 9 | IJGC 2015 gold medal | 18 | 14 | 22.3 | 1,482.2 | 5.9 | 5.7 | 5.0 | 5.6 | 5.5 | 4.8 | 5.42 ± 0.43 |
| 10 | Asian Games 2018 silver medal (All Japan 2018 gold medal) | 18 | 12 | 21.8 | 1,588.0 | 5.9 | 6.1 | 5.3 | 5.2 | 6.0 | 5.6 | 5.68 ± 0.38 |
| Mean ± SD | | 19.9 ± 1.4 | 13.6 ± 2.3 | 22.9 ± 1.0 | 1,506.5 ± 74.9 | 5.82 ± 0.27 | 5.86 ± 0.39 | 5.57 ± 0.33 | 5.24 ± 0.23 | 5.90 ± 0.30 | 5.55 ± 0.40 | |

*Note:* All WCGs in this study have won medals in gymnastics world championships since 2015.

Abbreviations: BMI, body mass index; D-score, difficulty score; DTB, Deutscher Turner-Bund; ICV, intracranial volume; IJGC, International Junior Gymnastics Competition; SD, standard deviation; WC, World Cup; WGC, World Gymnastics Championships.

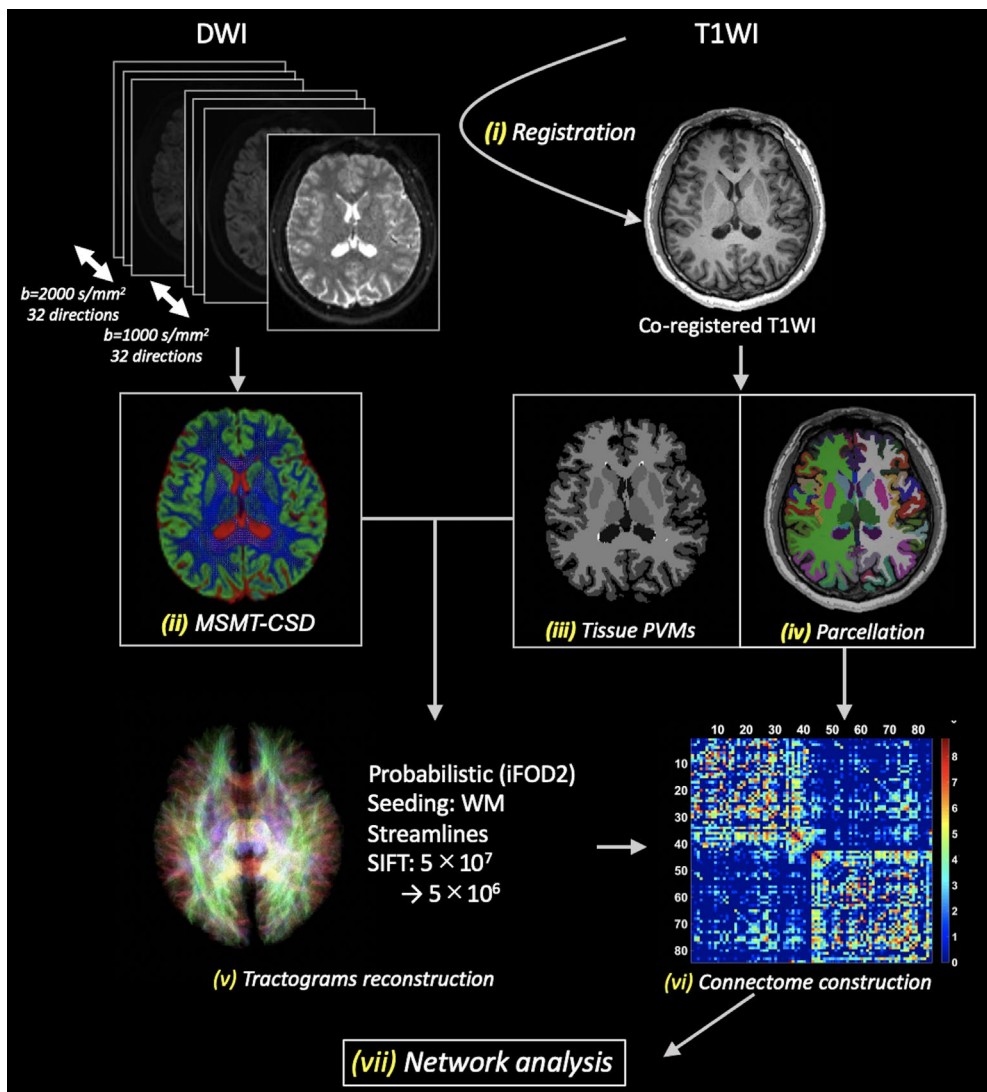

**FIGURE 1** Schematic of structural network mapping. Structural connectomes were mapped according to the following steps: (1) registration of T1WIs to the DWI, (2) estimation of FODs using MSMT-CSD, (3) estimation of tissue partial volume maps, (4) parcellation of cortical and subcortical GM, (5) reconstruction of streamline tractogram using MSMT-CSD, (6) construction of structural connectomes, and (7) network-based statistics and graph theoretical analysis. GM, gray matter; iFOD2, second-order integration over FOD algorithm; MSMT-CSD, multishell, multitissue constrained spherical deconvolution; WM, white matter

location and extent of the subcortical GM segmentations generated by FreeSurfer (Dale et al., 1999).

## 2.5 | Defining edges

MSMT-CSD probabilistic tracking with multishell DW-MRI data ($b$-values of 0, 1,000, and 2,000 s/mm²) utilizing the MRtrix3 software was used to acquire whole-brain tractograms.

Multiple response functions as functions of tissue type and b-value were calculated for MSMT-CSD (Jeurissen et al., 2014). Voxels were allocated to WM if the fractional anisotropy was >0.7 and the WM tissue probability based on the structural image was >0.95. Next, the DW-MRI signal was reorientated to correct the alignment of the principal axis of diffusion. Then, WM anisotropic

response functions for each shell were calculated by averaging the DW-MRI profiles over these voxels. Further, voxels were allocated to the GM and CSF if tissue segmentation had a volume fraction >95% and fractional anisotropy <0.2. DW-MRI profiles for each shell were averaged to acquire response functions for GM and CSF. Lastly, the *dwi2fod* command with the *msmt-csd* option in MRtrix3 was utilized to acquire fiber orientation distribution functions (fODFs) of WM, GM, and CSF. Maximum spherical harmonic orders of lmax were set to 6, 0, and 0 for WM, GM, and CSF, respectively.

The output WM-fODFs were used for probabilistic MSMT-CSD tracking, applying the second-order integration over the FOD (iFOD2) algorithm (Tournier et al., 2010), with parameters as follows: step size = 1.0 mm, maximum curvature = 45° per step, length = 4–200 mm, and fiber orientation distribution. Then, the spherical-deconvolution informed filtering of tractogram (SIFT) model (Smith

et al., 2013) was utilized to dynamically determine seed points. The reconstruction was filtered from $5 \times 10^7$ to $5 \times 10^6$ streamlines by also utilizing SIFT. Additionally, back-tracking was applied within the ACT framework (Smith et al., 2012).

## 2.6 | Constructing connectomes

A connectome modeled as an undirected and weighted network was reconstructed for each subject based on connectivity derived from the MSMT-CSD probabilistic tracking.

Nodes and the number of streamlines connecting each pair of nodes were allocated. Assigning streamlines to the closest node within a 2-mm radius of each streamline endpoint in ACT (Smith et al., 2015) produced an $84 \times 84$ interregional connectivity matrix. Each element in the matrix was filled with the number of streamlines, which provided an indicator of connectivity strength. Any self-connections, appearing as diagonal elements, were excluded from analysis. Spurious links were removed by applying a connection density threshold (T) (Rubinov & Sporns, 2010). Bias from any single threshold was minimized by examining global network metrics across a spectrum of thresholds ($10\% < T < 30\%$ in 5% increments) (Zhang et al., 2011). Lastly, the pairs of regions with the lowest streamline counts were allocated a value of zero, and regions in the top T% based on streamline count were not changed.

## 2.7 | Analyses using graph theory

Graph theory (Bullmore & Sporns, 2009) was utilized to calculate the brain anatomical network characteristics of WCGs. The Brain Connectivity Toolbox (http://www.brain-connectivity-toolbox.net/) was utilized to perform the topological measurements of connectivity matrices. As a result, five global (characteristic path length, clustering coefficient, global efficiency, mean strength, and small-worldness ratio) and five local (nodal strength, nodal degree, betweenness centrality, local clustering, and local efficiency) network metrics were produced. See the summary in Supporting Information (Table S1) for details of the global and local network metrics. The magnitude of effect of each global network metric between the WCG and control groups was calculated using Cohen's $d$, incorporating the wide range of thresholds to remove any spurious links ($10\% < T < 30\%$ in 5% increments). A threshold of 30% was utilized for both global and local network metrics as Cohen's $d$ was typically greatest at this threshold (Supporting Information Table S2).

## 2.8 | Detection of disrupted WM connectivity

Subnetworks (clusters of nodes and edges), including connections in WCGs (e.g., increased streamline count), were detected using network-based statistics (NBS) (see Zalesky et al., 2010 for details of NBS). An analysis of covariance (ANCOVA) was performed independently at each edge to test the null hypothesis of equivalence of the mean streamline count of WCGs and controls. Age, body mass index (BMI), and ICV were included as nuisance covariates for possible confounding factors in brain microstructural changes (Dekkers et al., 2019; Kijonka et al., 2020). A primary component-forming threshold (e.g., $p = 0.01$, $t = 2.55$, two-tailed $t$ test) was utilized to determine a set of suprathreshold edges (see Supporting Information Table S3 for results across different thresholds). Statistical significance with respect to an empirical calculation of the null distribution of maximal component sizes (10,000 permutations) was acquired for each component, with the number of edges in each component representing component size. If a component had a $p$-value of 0.05 following family-wise correction, it was reported. Neural nodes with significant components (subnetworks) were assigned following NBS analysis to five function systems, according to previous studies: sensorimotor, default mode, attentional, visual, and limbic/subcortical systems (He et al., 2009; Wang et al., 2013).

## 3 | STATISTICAL ANALYSIS

All statistical analyses were performed using IBM SPSS for Windows, version 22.0 (IBM Corp., Armonk, NY, USA). In accordance with the Kolmogorov–Smirnov test, between-group differences were analyzed by Student $t$-tests for age, BMI, and ICV and by chi-squared tests for gender. Moreover, the differences in global and local topological metrics were assessed using ANCOVA with age, BMI, and ICV as nuisance covariates. To correct for multiple comparisons, we used the false discovery rate, with a $p$-value $< 0.05$ considered significant.

In addition, we used the Pearson's correlation coefficient to test for relationships between brain measures (e.g., connectivity strength and topological metrics) that exhibited significant between-group differences and the D-score for each gymnastics event (floor exercise, horizontal bar, rings, vault, parallel bars, and pommel horse) or years of training.

## 4 | RESULTS

### 4.1 | Changes in WM connection characteristics of Japanese WCGs

We performed NBS analysis to detect changes in subnetwork connectivity that appeared to be specific to WCGs. The null hypothesis of equality in the mean streamline count between WCGs and controls was rejected ($p < 0.05$) for networks involving the sensorimotor, default mode, attentional, visual, and limbic areas (Figure 2; Supporting Information Table S4). Specifically, the NBS identified significantly increased subnetwork connectivity comprising 67 edges and 53 nodes in the WCG group relative to the control group ($p = 0.020$). We did not detect significantly decreased subnetwork connectivity in WCGs relative to controls.

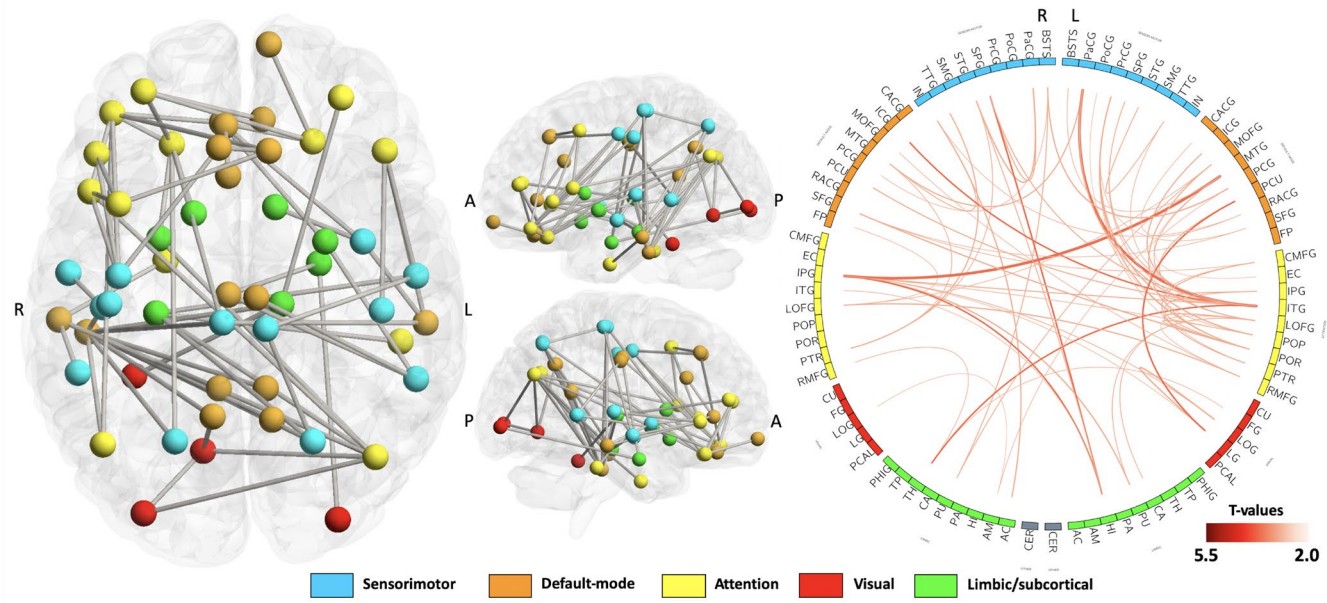

**FIGURE 2** Subnetworks with significant increased streamline count in world-class gymnasts relative to controls. The neural nodes comprising significant subnetworks were assigned to five functional systems—the sensorimotor, default mode, attentional, visual, and limbic/subcortical systems—color coded as blue, orange, yellow, red, and green, respectively. A, anterior; P, posterior; L, left hemisphere; R, right hemisphere

All of the edges with increased connectivity in WCGs were connected to brain areas that were classified as sensorimotor, attention, visual, limbic/subcortical, and default mode systems. These 19 edges can be classified into intrasystem connections and 48 intersystem connections. Among the intrasystem connections, eight edges were connected within attention areas, three edges were connected within the sensorimotor areas, seven edges were connected within default mode areas, and one edge was connected with visual area. For the intersystem connections, 33 of 48 edges were connected between the sensorimotor, attention, default mode and visual systems.

## 4.2 | Relationships between WM connection characteristics and D-score in WCGs

We detected significant correlations between the streamline count of some edges that showed significant between-group differences and the D-score for each event or years of training (Figure 3; Table 2). The D-score for floor was positively correlated with the streamline count of the edges linking between the left isthmus of the cingulate gyrus (default mode area) and left cerebellum cortex (other area). The D-score for parallel bars was positively correlated with the streamline count of the edges linking between the right medial orbitofrontal gyrus (default mode area) and right pars orbitalis of the inferior frontal gyrus (attention area), and between left middle temporal gyrus (default mode area) and left insula (sensorimotor area). The D-score for horizontal bar was positively correlated with the streamline count of the edges linking between the right medial orbitofrontal gyrus (default mode area) and right pars orbitalis of the

inferior frontal gyrus (attention area), and between the left precuneus and right inferior temporal gyrus (default mode area). We did not find a significant correlation between the D-score for other gymnastic events and the streamline count.

In addition, we found that the number of years of training was positively correlated with the streamline count of the edges linking between the left hippocampus (limbic/subcortical area) and right lateral occipital gyrus (visual area), and between the left inferior parietal gyrus (attention area) and right insula (sensorimotor area).

## 4.3 | Global metric characteristics of the WCGs

To clarify the characteristics of the whole-brain topological network of the WCGs, we conducted a statistical comparison of five global metrics (characteristic path length, clustering coefficient, global efficiency, mean strength, and small-worldness ratio) between the WCGs and controls. Among the five global metrics, we noted significant differences in global connectivity strength and characteristic path length between the two groups. Specifically, WCGs displayed significantly increased global connectivity strength as well as decreased characteristic path length (Table 3). In addition, although both WCGs and controls demonstrated small-world organization ($\sigma > 1$), no between-group difference was found in the small-worldness ratio. Furthermore, there were no significant differences in the clustering coefficient or global efficiency between the two groups.

No correlations were detected between the global network metrics and the D-score for each event or for years of training in the WCG group.

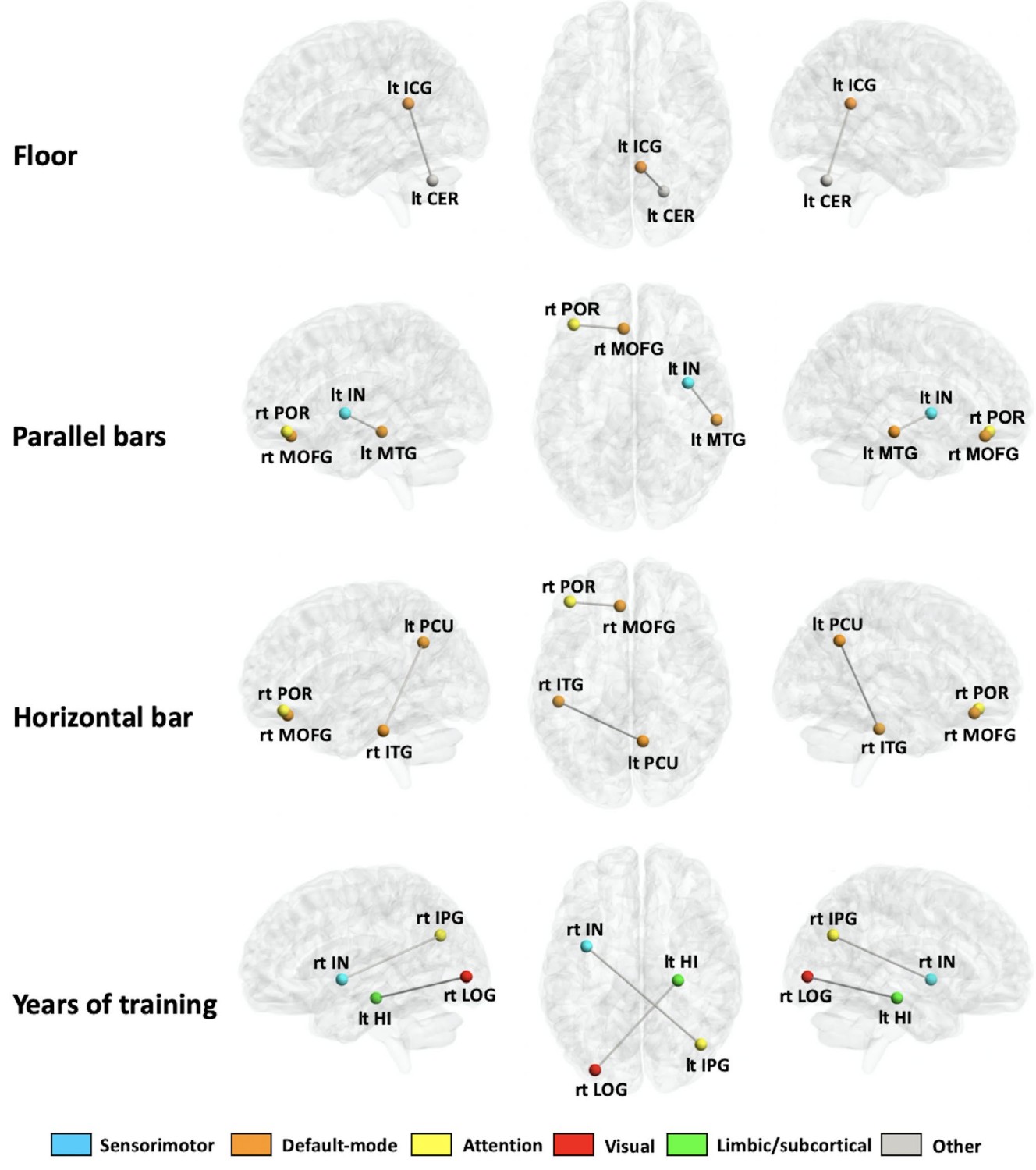

**FIGURE 3** Subnetworks with a significant correlation between streamline count and absolute D-score for each gymnastics event or years of training in world-class gymnasts. IPG, inferior parietal gyrus; ITG, inferior temporal gyrus; LOFG, lateral orbitofrontal gyrus; MOFG, medial orbitofrontal gyrus; POR, pars orbitalis; PrCG, precentral gyrus; RACG, rostral anterior cingulate; RMFG, rostral middle frontal gyrus; SFG, superior frontal gyrus; TP, temporal pole

## 4.4 | Local metric characteristics of WCGs

To identify the characteristics of each brain region (node) in the WCGs, we performed a statistical comparison of the five local metrics (nodal strength, nodal degree, betweenness centrality, local clustering, and local efficiency) between WCGs and controls. No between-group differences in local topological metrics were observed after adjusting for age, BMI, and ICV. However, with only

age included as a covariate, 12 brain regions with significantly higher nodal strength and two brain regions with significantly higher nodal degree were detected in the WCGs compared with the controls (Figure 4; Table 4). Specifically, the 12 brain areas that exhibited increased nodal strength can be classified into two sensorimotor nodes (bilateral precentral gyrus), three default mode nodes (right superior frontal gyrus, right rostral anterior cingulate gyrus, and right medial orbitofrontal gyrus), five attentional nodes (right lateral orbitofrontal gyrus, right rostral middle frontal gyrus, right pars orbitalis of inferior frontal gyrus, right inferior temporal gyrus, and right inferior parietal gyrus), two limbic/subcortical nodes (bilateral temporal pole). In addition, the two brain areas that exhibited an increased nodal degree were both attentional nodes (bilateral inferior parietal lobule).

**TABLE 2** Correlation analysis for the relationships between mean streamline counts of the edges exhibited significant differences as detected by network-based statistics and difficulty-scores of each gymnastic event, years of training

| Connection | p-value | R |
|---|---|---|
| *Floor* | | |
| Left isthmus cingulate to left cerebellum cortex | 0.006 | 0.739 |
| *Parallel bars* | | |
| Right medial orbitofrontal cortex to right pars orbitalis | 0.015 | 0.739 |
| Left middle temporal gyrus to left insula cortex | 0.042 | 0.649 |
| *Horizontal bar* | | |
| Right medial orbitofrontal cortex to right pars orbitalis | 0.039 | 0.658 |
| Left precuneus to right inferior temporal gyrus | 0.043 | 0.648 |
| *Years of training* | | |
| Left hippocampus to right lateral occipital gyrus | 0.023 | 0.501 |
| Left inferior parietal gyrus to right insula cortex | 0.040 | 0.507 |

We did not find any correlations between the local network metrics and D-scores for each event or for the number of years of training in the WCG group.

## 5 | DISCUSSION

To clarify how the brain network structure of gymnasts differs from that of nonathletes, in this study we compared DW-MRI-based connectomes derived from probabilistic MSMT-CSD tracking between Japanese WCGs and controls. Using NBS, we identified increased connection density of the subnetworks involving the sensorimotor, default mode, attentional, visual, and limbic areas in WCGs. The connection density of these subnetworks was correlated with D-scores of some gymnastic events or years of training. In addition, graph theory analysis revealed the characteristics of brain anatomical topology in the WCGs. At the global level, the WCGs displayed significantly increased global connectivity strength as well as decreased characteristic path length in comparison with controls. At the local level, the WCGs showed higher nodal strength and degree in the sensorimotor, default mode, attention, and limbic/subcortical areas as compared with controls.

### 5.1 | Changes in the WM connection characteristics of the WCGs

We found that all 67 edges with increased connectivity in WCGs were connected to brain regions that were classified to the sensorimotor, attention, visual, limbic/subcortical, and default mode systems. Specifically, the edges consisted of 19 intrasystem connections and 48 intersystem connections. Interestingly, all 19 intrasystem connections are within the sensorimotor, attention, visual, and default mode system, and most of the 54 intersystem

**TABLE 3** Between-group comparison of global network measures

| | Controls | World-class gymnasts | F | p-value[*] | Partial eta-squared |
|---|---|---|---|---|---|
| Global clustering (*SD*) | 0.0051 (0.0007) | 0.0049 (0.0008) | 0.034 | 0.86 | 0.0023 |
| Global efficiency (*SD*) | 0.040 (0.006) | 0.041 (0.007) | 0.86 | 0.43 | 0.054 |
| Global strength (*SD*) | 4,751.48 (295.92) | 5,205.66 (329.46) | 18.40 | 0.0050 | 0.55 |
| Characteristic path length (*SD*) | 0.0064 (0.0004) | 0.0057 (0.0005) | 13.44 | 0.0080 | 0.47 |
| Small-worldness ratio (*SD*) | 2.34 (0.44) | 2.62 (0.56) | 2.82 | 0.16 | 0.16 |

*Note:* Data are expressed as mean (*SD*).

Abbreviation: *SD*, standard deviation.

*False discovery rate-corrected *p*-values.

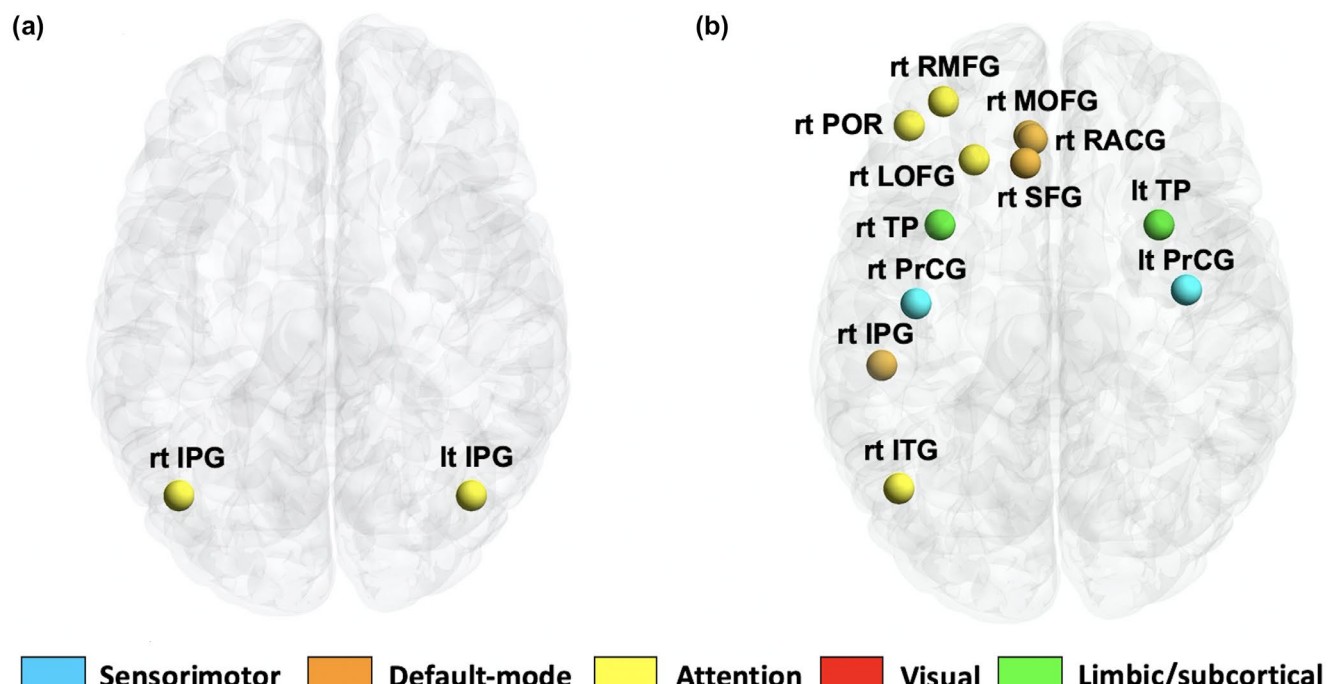

**FIGURE 4** Local metric results. Regions with a significantly ($p < 0.05$, false discovery rate corrected) increased nodal degree (a) and nodal strength (b) in world-class gymnasts as compared with controls. lh, left hemisphere; rh, right hemisphere; IPG, inferior parietal gyrus; ITG, inferior temporal gyrus; LOFG, lateral orbitofrontal gyrus; MOFG, medial orbitofrontal gyrus; POR, pars orbitalis; PrCG, precentral gyrus; RACG, rostral anterior cingulate; RMFG, rostral middle frontal gyrus; SFG, superior frontal gyrus; TP, temporal pole

connections (33/48) link the sensorimotor, attention, visual, and default mode systems. The sensorimotor system plays an important role in the acquisition and performance of motor skills (Arce-McShane et al., 2016), the default mode system is implicated in automated information processing (Vatansever et al., 2017), and the attention system is responsible for attention processing (Petersen & Posner, 2012). The functions for which each system is responsible are all important functions in gymnastics, and the results of the NBS analysis support the evidence that these networks underlie the extraordinary motor skills of WCGs. In another study using NBS, the authors reported an increased connectivity density between the sensorimotor, default mode, attention, and limbic/subcortical areas in WCGs (Wang et al., 2013), which is also in agreement with our results. Furthermore, the fact that the structural connectivity in some of the connections was significantly associated with years of training suggests that these changes in brain connections are formed by prolonged training.

## 5.2 | Relationships between WM connection characteristics and D-score in the WCGs

This study also revealed significant associations between the increased structural connectivity of some brain structures related to the gymnastic ability and the D-score, which is used as an index of the gymnasts' specific physical skills for each apparatus.

For floor exercise, perceptual systems that control body orientation and angular velocity during flight are generally important (Jemni, 2011). The D-score of floor routines was positively correlated with the structural connectivity between the left isthmus of the cingulate gyrus and left cerebellum cortex. The isthmus of the cingulate gyrus and the cerebellum is related to spatial navigation (Vann et al., 2009) and fine movement, equilibrium, posture, and motor learning in humans (Fine et al., 2002), respectively. The functions governed by these regions are necessary for controlling body orientation and angular velocity during flight, and this correlation suggests that the strength of the connections between these regions might be the neural basis of floor movement.

In the parallel bars and horizontal bar routines, the mechanics of the giant swings and takeoff requirements for dismounts and release–regrasp techniques are important factors (Jemni, 2011). It is presumed that the ability to perform giant swings and takeoffs requires a high level of spatial perception ability and visual function, and release–regrasp techniques require a high level of visual function and hand perception ability. The D-score for the parallel bars was positively correlated with the structural connectivity between the right medial orbitofrontal gyrus and right pars orbitalis of the inferior frontal gyrus, and between the left middle temporal gyrus and left insula. The middle temporal gyrus was closely related to visual motion perception (Gao et al., 2020), and the insula was associated with socioemotional and cognitive function as well as with sensorimotor processing, including hand

**TABLE 4** Regions with a significant between-group difference in nodal strength and nodal degree

| Regions | Controls | WCG | t | p-value* | Cohen's d |
|---|---|---|---|---|---|
| *Nodal strength* | | | | | |
| Right lateral orbitofrontal gyrus | 3,113.6 | 3,882.2 | 4.64 | 0.002 | 2.19 |
| | (251.28) | (429.13) | | | |
| Right temporal pole | 866.9 | 1,165.6 | 4.07 | 0.007 | 1.91 |
| | (115.62) | (187.53) | | | |
| Right rostral middle frontal gyrus | 7,007.6 | 8,596.1 | 3.84 | 0.001 | 1.81 |
| | (883.57) | (873.01) | | | |
| Right pars orbitalis | 1,892.5 | 2,390.8 | 3.79 | 0.001 | 1.79 |
| | (229.18) | (320.96) | | | |
| Left temporal pole | 1,026.2 | 1,437.1 | 3.66 | 0.002 | 1.72 |
| | (257.68) | (217.08) | | | |
| Right precentral gyrus | 10,899.3 | 12,479.5 | 3.46 | 0.003 | 1.63 |
| | (919.52) | (1,014.8) | | | |
| Right inferior temporal gyrus | 5,175.1 | 6,248.9 | 3.31 | 0.004 | 1.56 |
| | (758.2) | (611.2) | | | |
| Right inferior parietal gyrus | 8,250.1 | 9,561.2 | 3.21 | 0.005 | 1.51 |
| | (590.8) | (1,074.4) | | | |
| Right superior frontal gyrus | 13,275 | 14,833.8 | 3.19 | 0.005 | 1.50 |
| | (938.3) | (1,128.5) | | | |
| Right rostral anterior cingulate gyrus | 1,646.7 | 1,969.9 | 3.18 | 0.005 | 1.50w |
| | (207.4) | (223.1) | | | |
| Right medial orbitofrontal gyrus | 2,209.1 | 2,566.9 | 3.12 | 0.006 | 1.47 |
| | (253.9) | (232.4) | | | |
| Left precentral gyrus | 11,589.9 | 13,222.2 | 3.04 | 0.006 | 1.44 |
| | (1,151.3) | (1,123.2) | | | |
| *Nodal degree* | | | | | |
| Left inferior parietal gyrus | 55.7 | 62.2 | 4.71 | 0.002 | 2.22 |
| | (3.38) | (2.4) | | | |
| Right inferior parietal gyrus | 58.2 | 63.1 | 4.31 | >0.001 | 2.03 |
| | (2.44) | (2.39) | | | |

*Note:* Data are expressed as mean (*SD*).

Abbreviations: *SD*, standard deviation; WCG, world-class gymnast.

*False discovery rate-corrected *p*-values.

perception (Uddin et al., 2017). Therefore, all of these areas are related to spatial recognition, vision, and hand perception, which are necessary for parallel bar performance. The D-score for the horizontal bar was positively correlated with the structural connectivity between the right medial orbitofrontal gyrus and right pars orbitalis of the inferior frontal gyrus, and between the left precuneus and right inferior temporal gyrus. The precuneus is responsible for visuospatial imagery, episodic memory, and consciousness (Cavanna & Trimble, 2006), and the inferior temporal gyrus is associated with recognition/identification of objects in the field of view (Lafer-Sousa & Conway, 2013); thus, the connection

between these structures can be considered essential for the horizontal bar. Further, the structural connectivity between the right medial orbital frontal gyrus and the inferior frontal gyrus was associated with the ability of both parallel bars and horizontal bar. The medial orbitofrontal gyrus and inferior frontal gyrus pars orbitalis are related to memory, decision-making (Euston et al., 2012), and language processing (Wiegell et al., 2000). The ability to memorize the routines and decision-making skills is important for WCGs; however, it is difficult to interpret the relationship between the language-processing function of the pars orbitalis of the inferior frontal gyrus with the parallel bars and horizontal bar.

## 5.3 | Global and local metric characteristics of the WCGs

In graph theory analysis, the WCGs displayed significantly increased global connectivity strength as well as decreased characteristic path length as compared with controls. Increased global strength indicates overall strong structural connectivity in the brain, and reduced characteristic path lengths suggest efficient network integration (i.e., high capability of parallel information transfer). The changes in global metrics seen in gymnasts might reflect the neuroplasticity of the brain's structural connectivity, which results in more efficient and faster perception, object recognition, situation awareness, decision-making, and motor control processes. Indeed, to maximize their performance within a short competition time, gymnasts need these abilities. In line with our study, Wang et al. (2013) also reported a decreased characteristic path length in gymnasts. However, in contrast to their study, we did not find a significantly increased global efficiency in WCGs. The discrepancy between our results and those of the previous authors might be due not only to the different tractography generation algorithms (as described below) used to map the connectome but also to the different ability levels of the gymnasts (world championships medalists in this study and Olympic gold medalists in the study by Wang et al.). It is generally believed that Olympic gold medalists have greater gymnastics abilities than world championship winners.

At the local level, after adjusting for age, WCGs show increased "functional importance," as indicated by increased nodal strength in the sensorimotor, default mode, attention, and limbic/subcortical system compared with controls. As mentioned previously, the sensorimotor system plays an important role in the acquisition and performance of motor skills (Arce-McShane et al., 2016), the default mode system is implicated in automated information processing (Vatansever et al., 2017), and the attention system is responsible for attention processing (Petersen & Posner, 2012). In addition, the limbic/subcortical system is related to emotion, cognition, fear, and motivation (Salzman & Fusi, 2010). Wang et al., (2013) reported an increase in the functional importance of the motor and attention system, which is partially consistent with our findings. We also found that the functional importance of the default mode was associated with automated information processing and that the limbic/subcortical system was related to emotion, cognition, fear, and motivation. The automation of information processing is a necessary function for performing complicated aerial movements in quick succession, and controlling emotion, cognition, fear, and motivation is important for achieving the best performance under the high tension of a world championship. Therefore, the changes in the brain topological properties of the WCGs, characterized by the increased functional importance of the areas governing these functions, are very reasonable. In this study, the WCGs also showed higher functional segregation (i.e., increased special information processing), as indicated by the increased nodal degree in the bilateral inferior parietal gyrus (attention area). As mentioned before, the inferior parietal gyrus is associated with spatial perception and visuomotor integration (Andersen, 2011); therefore, topological changes characterized by increased segregation in the bilateral inferior parietal gyrus are a rational change.

## 5.4 | Methodological aspects

In this study, we used probabilistic MSMT-CSD tracking for connectome mapping. A previous study evaluated the connectome of gymnasts by deterministic tracking based on the fiber assignment by continuous tracking algorithm (Wang et al., 2013); however, that method cannot accurately estimate the neural fiber connections in regions with crossing and kissing fibers at the voxel level (Mori et al., 2004). The inability to resolve multiple fiber orientations in regions of crossing or kissing fibers, which include up to 90% of the WM voxels in the brain, may contribute to an unreliable estimation of connectome mapping in these regions (Douaud et al., 2011). In contrast, we applied probabilistic tracking, which can deal with the crossing/kissing problem, and therefore obtained more accurate estimates of the connectome. In addition, we adopted the MSMT-CSD method, which enables precise fODF estimates at the GM–WM interface (Jeurissen et al., 2014). These accurate estimates are essential for accurately characterizing both nodes (GM regions) and the edges between nodes. However, an important drawback of probabilistic tractography is that its high sensitivity typically comes at the expense of low specificity. In other words, it is important to note that probabilistic tracking is more likely to generate false fibers (false-positives) (Knosche et al., 2015; Thomas et al., 2014). Spurious fiber estimates are particularly detrimental to the characterization of connectome topological properties (Zalesky et al., 2016).

## 5.5 | Limitations

This study has some limitations: first, the sample size for this study was small (10 WCGs and 10 controls) because of the very limited number of active Japanese WCGs. In addition, an additional longitudinal study with a larger sample size is needed to determine whether genetic or early-life environmental factors influence the brain network topology of WCGs and/or whether long-term intensive training is a major contributor to the changes. In addition, because this study targeted only male WCGs, a future study evaluating female WCGs is necessary. It is also worth noting that brain anatomical network plasticity shown in this study might not be specific to WCGs. Further studies should also evaluate brain network organization in world-class athletes with different sports.

## 6 | CONCLUSION

To clarify how the brain network structure of gymnasts is different from that of controls, we mapped the connectomes of WCGs using DW-MRI–based connectomes derived from probabilistic MSMT-CSD tracking. As a result, we found an increased connection density of the subnetworks involving the sensorimotor, default mode, attentional, visual, and limbic areas in WCGs. The density in some of the connections that constitute these subnetworks was correlated with years of training and the D-score for each gymnastic event. In addition, graph theory analysis revealed that the anatomical

topology of the gymnast's brain is characterized by a significantly increased global connection strength and a decreased characteristic path length (i.e., high parallel information transfer capability). These changes in brain topology may represent the neural basis for the outstanding gymnastics performance that results from brain plasticity. The indices of brain topology (i.e., connection density and graph theory indices) could become markers for the objective evaluation of gymnastic performance.

## DECLARATION OF TRANSPARENCY

The authors, reviewers and editors affirm that in accordance to the policies set by the *Journal of Neuroscience Research*, this manuscript presents an accurate and transparent account of the study being reported and that all critical details describing the methods and results are present.

## ACKNOWLEDGMENTS

The authors are grateful to all participants for their contribution in this study.

## CONFLICT OF INTEREST

The authors declare no competing interests.

## AUTHOR CONTRIBUTIONS

Conceptualization, H.T., K.K., C.A., H.W., H.S., M.H., S.A., and H.N.; Data Curation, H.T., K.K., C.A., W.U., M.F., Y.T., T.M., M.L., A.H., S.F., A.W., T.A., and S.M.; Formal Analysis, H.T., K.K., and C.A.; Investigation, H.T., K.K., and C.A.; Methodology, H.T., K.K., and C.A.; Writing – Original Draft, H.T., K.K.; Writing – Review & Editing, H.T., K.K., C.A., W.U., M.F., H.W., H.S., Y.T., T.M., M.L., A.H., S.F., A.W., T.A., S.M., M. H., S.A., and H.N.; Funding Acquisition, Y.T. and S.A.; Supervision, K.K., C.A., S.A., and H.N.; Validation, H.T., K.K., and C.A.; Software, W.U.; Visualization, W.U.

## PEER REVIEW

The peer review history for this article is available at https://publons.com/publon/10.1002/jnr.24912.

## DATA AVAILABILITY STATEMENT

The data that support the findings of this study are available from the corresponding author upon reasonable request.

## ORCID

*Christina Andica*  https://orcid.org/0000-0002-9339-6950

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

## SUPPORTING INFORMATION

Additional Supporting Information may be found online in the Supporting Information section.

Transparent Science Questionnaire for Authors

Transparent Peer Review Report

**TABLE S1** Interpretations of graph metrics

**TABLE S2** Cohen's *d* of each global metric across the full range of sparsity thresholds for comparison between the world-class gymnasts and controls

**TABLE S3** Networks identified as significantly different between world-class gymnasts and controls using network-based statistical analysis

**TABLE S4** Network identified as significantly different between world-class gymnasts and controls using network-based statistical analysis

