## [Transparent Peer Review Report · Journal of Neuroscience Research]

**Connectome analysis of male world-class gymnasts using probabilistic multishell, multitissue
constrained spherical deconvolution tracking**

**Kamagata, Koji; Tomita, Hiroyuki; Andica, Christina; Uchida, Wataru; Fukuo, Makoto; Waki, Hidefumi;
Sugano, Hidenori; Tange, Yuichi; Mitsuhashi, Takumi; Lukies, Matthew; Hagiwara, Akifumi; Fujita,
Shohei; Wada, Akihiko; Akashi, Toshiaki; Murata, Syo; Harada, Mutsumi; Aoki, Shigeki; Naito, Hisashi**

Review timeline:

Submission date: 12 January 2021

Editorial Decision: Minor Modification (24 March 2021)

Revision Received: 19 April 2021

Accepted: 17 June 2021

Editor 1: Sandra Chanraud

Editor 2: David McArthur

Editor 3: Junie Warrington

Reviewer 1: Ryan Cabeen

1st Editorial Decision

Decision letter

Dear Dr Kamagata:

Thank you for submitting your manuscript to the Journal of Neuroscience Research. We have now received the reviewer feedback and have appended those reviews below. I am glad to say that the reviewers are overall very enthusiastic and supportive of the study. They did raise some concerns and made some suggestions for clarification, but I expect that these points should be relatively straightforward to address. If there are any questions or points that are problematic, please feel free to contact me. I will be glad to discuss.

We ask that you return your manuscript within 30 days. Please explain in your cover letter how you have changed the present version and submit a point-by-point response to the editors' and reviewers' comments. If you require longer than 30 days to make the revisions, please contact Dr Junie Warrington (jpwarrington@umc.edu). To submit your revised manuscript: Log in by clicking on the link below <https://wiley.atyponrex.com/submissionBoard/1/6d252ab8-d78f-4e32-8e2a-24e90476e6ba/current>

(If the above link space is blank, it is because you submitted your original manuscript through our old submission site. Therefore, to return your revision, please go to our new submission site here (submission.wiley.com/jnr) and submit your revision as a new manuscript; answer yes to the question "Are you returning a revision for a manuscript originally submitted to our former submission site (ScholarOne Manuscripts)? If you indicate yes, please enter your original manuscript's Manuscript ID number in the space below" and including your original submission's Manuscript ID number (jnr-2021-Jan-9438) where indicated. This will help us to link your revision to your original submission.)

The journal has adopted the "Expects Data" data sharing policy, which states that all original articles and reviews must include a Data Availability Statement (DAS). Please see <https://authorservices.wiley.com/author-resources/Journal-Authors/open-access/data-sharing-citation/data-sharing-policy.html#standardtemplates> for examples of an appropriate DAS. Please include the DAS in the manuscript as well.

Thank you again for your submission to the Journal of Neuroscience Research; we look forward to reading your revised manuscript.

Best Wishes,

Dr Sandra Chanraud

Associate Editor, Journal of Neuroscience Research

Dr Junie Warrington
Editor-in-Chief, Journal of Neuroscience Research

Editorial Comments to the Author:

1. Please upload a graphical abstract, which we are asking of all authors submitting original research articles. This is intended to provide readers with a visual representation of the conclusions and an additional way to access the contents and appreciate the main message of the work. What we require is a .tif image file and a .doc text file containing an abbreviated abstract. For the image, labels, although useful, must be kept to a minimum and the image should be 400 x 300, 300 x 400, or 400 x 400 pixels square and at a resolution of 72 dpi. This can be one of the figures from your article, or something slightly different, as long as it represents your study. Instructions for this can be found in our author guidelines online at [http://onlinelibrary.wiley.com/journal/10.1002/\(ISSN\)1097-4547/homepage/ForAuthors.html](http://onlinelibrary.wiley.com/journal/10.1002/(ISSN)1097-4547/homepage/ForAuthors.html)
2. The National Institutes of Health now mandates the inclusion of sex as a biological variable. To conform with this mandate, the Journal of Neuroscience Research has established new policy (please see our editorial: <http://onlinelibrary.wiley.com/doi/10.1002/jnr.23979/full>) requiring all authors to ensure proper consideration of sex as a biological variable. Please ensure that: 1) Any paper utilizing subjects of one sex state the sex of the sample in the title and abstract; 2) The number of samples/subjects of each sex used in the research must be clearly stated in the methods section; 3) The inability for any reason to study sex differences where they may exist should be discussed as a study limitation. We are also encouraging authors to report exploratory analyses of potential sex differences in studies not explicitly designed to address them
3. Please add to your paper (after the Discussion and Acknowledgments, immediately before the References) a conflict of interest statement and a statement of authors' contributions. The statement must follow the CRediT Taxonomy. You can find examples of such statements in the author guidelines on-line at [http://onlinelibrary.wiley.com/journal/10.1002/\(ISSN\)1097-4547/homepage/ForAuthors.html](http://onlinelibrary.wiley.com/journal/10.1002/(ISSN)1097-4547/homepage/ForAuthors.html)
4. Authors must submit, in the main text of the document, a 100-word-maximum statement about the significance of their research paper written at a level that is understandable to the general public and to scientists outside their field of specialty. This statement will be distinct in purpose from the abstract, with the primary goal of broadly explaining the relevance and importance of this work and how this work contributes to different diseases.
5. Please consider changing the orientation of Figure 3. The panels are too small to read the details. It may be best to display 3 brains per row instead of 6. Please also increase the font size used in each figure/panel.

Associate Editor: Chanraud, Sandra
Comments to the Author:
(There are no comments.)

Statistics Editor: McArthur, David
Comments to the Author:

MRtrix software is cited as available at brain.org.au but that website indicates that this version [0.2] "is no longer under active development and now only receives bug fix updates. The new MRtrix3 release, which includes the latest developments, is available on the MRtrix3 Website, along with its documentation." Presumably you are referring to version 0.2.

Table 4 shows a footnote "* Statistical significance" that appears to be unused and would not seem to be needed. The final p-value in that table is displayed to 4 decimal digits, which should read ">0.001" in keeping with all the other values (i.e., a stable degree of decimal exactitude).

Table S4 includes several t-value entries with a single decimal digit; these should be expressed as 2 digits in keeping with the other such values. The phrase "Identified connection (P = 0.01)" should match the narrative's value in Section 4.1 of "P = 0.011".

Reviewer: 1

Comments to the Author

This manuscript presents work investigating structural brain differences between world-class gymnasts and matched controls, as assessed by magnetic resonance imaging. In particular, the study focuses on network-level effects derived from tractography-based modeling of brain connectivity between cortical brain areas. Overall, I found the paper to be interesting and well-written, and the technical aspects of the image analysis reflect the state-of-the-art. I have several concerns to note:

Major Point 1: The manuscript makes several statements regarding neural plasticity and long-term intensive training, but I think the paper should more carefully note the limitations of the present study design. Specifically, because of the cross-sectional nature of the study, the data cannot tell us whether the brain differences are due to training, genetics, early life environment, etc. The discussion points out this limitation, but the abstract may lead the reader to think otherwise, so I think it would improve the manuscript to soften some of these claims. I think one other issue to consider in the limitations section is that the data may not indicate brain differences that are specific to gymnasts. It is plausible that other world-class athletes in other sports may have similar brain differences compared to controls, and the study design does not permit this aspect to be discerned.

Major Point 2: There are several other individual characteristics to consider in matching controls, beyond age and sex. In particular, due to high level of physical fitness of world-class athletes, it is important to consider body-mass index (BMI) as a potential confound, because it can have a significant impact on white matter when imaged using diffusion MRI [1]. The other factor to consider is head size. Because fiber count was an outcome of the results, it is important to ensure that, for whatever reason, the gymnasts didn't have different intra-cranial volume. It is common practice to perform multiple linear regression with these two factors as covariates, and I would strongly recommend the authors consider expanding their statistical analysis to implement this approach, or something to the same effect.

Minor Issues:

* On line 24: it would be helpful to have a reference supporting the point about genetic predisposition. This could be an important consideration about causal nature of the effects.

* On line 31: extra left parenthesis.

* It would be good to include references to relevant manuscripts for the MRtrix tools.

* It may be good to cite the recent evaluation study by Schilling et al. [2] which discusses tractography limitations that might impact structural connectome analysis.

[1] Dekkers, I. A., Jansen, P. R., & Lamb, H. J. (2019). Obesity, brain volume, and white matter microstructure at MRI: a cross-sectional UK Biobank study. *Radiology*, 291(3), 763-771.

[2] Schilling, K. G., Nath, V., Hansen, C., Parvathaneni, P., Blaber, J., Gao, Y., ... & Landman, B. A. (2019). Limits to anatomical accuracy of diffusion tractography using modern approaches. *NeuroImage*, 185, 1-11.

Authors' Response

Connectome analysis of male world-class gymnasts using probabilistic multishell, multitissue constrained spherical deconvolution tracking

Hiroyuki Tomita^{1*}, Koji Kamagata^{2*}, Christina Andica², Wataru Uchida², Makoto Fukuo², Hidefumi Waki², Hidenori Sugano³, Yuichi Tange³, Takumi Mitsuhashi³, Matthew Lukies⁴, Akifumi Hagiwara², Shohei Fujita², Akihiko Wada², Toshiaki Akashi², Syo Murata², Mutsumi Harada¹, Shigeki Aoki², Hisashi Naito¹

Review timeline: Submission date: January 12, 2021

Editorial Decision: March 24, 2021

Revision Received:

Editorial Decision:

Revision Received:

Accepted:

Editor:

Reviewer 1:

Reviewer 2:

1st Editorial Decision

24-Mar-2021

Dear Dr Kamagata:

Thank you for submitting your manuscript to the Journal of Neuroscience Research. We have now received the reviewer feedback and have appended those reviews below. I am glad to say that the reviewers are overall very enthusiastic and supportive of the study. They did raise some concerns and made some suggestions for clarification, but I expect that these points should be relatively straightforward to address. If there are any questions or points that are problematic, please feel free to contact me. I will be glad to discuss.

We ask that you return your manuscript within 30 days. Please explain in your cover letter how you have changed the present version and submit a point-by-point response to the editors' and reviewers' comments. If you require longer than 30 days to make the revisions, please contact Dr Junie Warrington (jpwarrington@umc.edu). To submit your revised manuscript: Log in by clicking on the link below <https://wiley.atyponrex.com/submissionBoard/1/6d252ab8-d78f-4e32-8e2a-24e90476e6ba/current> (If the above link space is blank, it is because you submitted your original manuscript through our old submission site. Therefore, to return your revision, please go to our new submission site here (submission.wiley.com/jnr) and submit your revision as a new manuscript; answer yes to the question "Are you returning a revision for a manuscript originally submitted to our former submission site (ScholarOne Manuscripts)? If you indicate yes, please enter your original manuscript's Manuscript ID number in the space below" and including your original submission's Manuscript ID number (jnr-2021-Jan-9438) where indicated. This will help us to link your revision to your original submission.)

The journal has adopted the "Expects Data" data sharing policy, which states that all original articles and reviews must include a Data Availability Statement (DAS). Please see <https://authorservices.wiley.com/author-resources/Journal-Authors/open-access/data-sharingcitation/data-sharing-policy.html#standardtemplates> for examples of an appropriate DAS. Please include the DAS in the manuscript as well.

Thank you again for your submission to the Journal of Neuroscience Research; we look forward to reading your revised manuscript.

Best Wishes,

Dr Sandra Chanraud

Associate Editor, Journal of Neuroscience Research

Dr Junie Warrington

Editor-in-Chief, Journal of Neuroscience Research

Editorial Comments to the Author:

1. Please upload a graphical abstract, which we are asking of all authors submitting original research articles. This is intended to provide readers with a visual representation of the conclusions and an additional way to access the contents and appreciate the main message of the work. What we require is a .tif image file and a .doc text file containing an abbreviated abstract. For the image, labels, although useful, must be kept to a minimum and the image should be 400 x 300, 300 x 400, or 400 x 400 pixels square and at a resolution of 72 dpi. This can be one of the figures from your article, or something slightly different, as long as it represents your study. Instructions for this can be found in our author guidelines online at [http://onlinelibrary.wiley.com/journal/10.1002/\(ISSN\)1097-4547/homepage/ForAuthors.html](http://onlinelibrary.wiley.com/journal/10.1002/(ISSN)1097-4547/homepage/ForAuthors.html)
2. The National Institutes of Health now mandates the inclusion of sex as a biological variable. To conform with this mandate, the Journal of Neuroscience Research has established new policy (please see our editorial: <http://onlinelibrary.wiley.com/doi/10.1002/jnr.23979/full>) requiring all authors to ensure proper consideration of sex as a biological variable. Please ensure that: 1) Any paper utilizing subjects of one sex state the sex of the sample in the title and abstract; 2) The number of samples/subjects of each sex used in the research must be clearly stated in the methods section; 3) The inability for any reason to study sex differences where they may exist should be discussed as a study limitation. We are also encouraging authors to report exploratory analyses of potential sex differences in studies not explicitly designed to address them
3. Please add to your paper (after the Discussion and Acknowledgments, immediately before the References) a conflict of interest statement and a statement of authors' contributions. The statement must follow the CRediT Taxonomy. You can find examples of such statements in the author guidelines on-line at [http://onlinelibrary.wiley.com/journal/10.1002/\(ISSN\)1097-4547/homepage/ForAuthors.html](http://onlinelibrary.wiley.com/journal/10.1002/(ISSN)1097-4547/homepage/ForAuthors.html)
4. Authors must submit, in the main text of the document, a 100-word-maximum statement about the significance of their research paper written at a level that is understandable to the general public and to scientists outside their field of specialty. This statement will be distinct in purpose from the abstract, with the primary goal of broadly explaining the relevance and importance of this work and how this work contributes to different diseases.
5. Please consider changing the orientation of Figure 3. The panels are too small to read the details. It may be best to display 3 brains per row instead of 6. Please also increase the font size used in each figure/panel.

Associate Editor: Chanraud, Sandra

Comments to the Author:

(There are no comments.)

Statistics Editor: McArthur, David

Comments to the Author:

MRtrix software is cited as available at brain.org.au but that website indicates that this version [0.2] "is no longer under active development and now only receives bug fix updates. The new MRtrix3 release, which includes the latest developments, is available on the MRtrix3 Website, along with its documentation." Presumably you are referring to version 0.2.

Table 4 shows a footnote "* Statistical significance" that appears to be unused and would not seem to be needed. The final p-value in that table is displayed to 4 decimal digits, which should read ">0.001" in keeping with all the other values (i.e., a stable degree of decimal exactitude).

Table S4 includes several t-value entries with a single decimal digit; these should be expressed as 2 digits in keeping with the other such values. The phrase "Identified connection (P = 0.01)" should match the narrative's value in Section 4.1 of "P = 0.011".

Reviewer: 1

Comments to the Author

This manuscript presents work investigating structural brain differences between world-class gymnasts and matched controls, as assessed by magnetic

resonance imaging. In particular, the study focuses on network-level effects derived from tractography-based modeling of brain connectivity between cortical brain areas. Overall, I found the paper to be interesting and well-written, and the technical aspects of the image analysis reflect the state-of-the-art. I have several concerns to note:

Major Point 1: The manuscript makes several statements regarding neural plasticity and long-term intensive training, but I think the paper should more carefully note the limitations of the present study design. Specifically, because of the cross-sectional nature of the study, the data cannot tell us whether the brain differences are due to training, genetics, early life environment, etc. The discussion points out this limitation, but the abstract may lead the reader to think otherwise, so I think it would improve the manuscript to soften some of these claims. I think one other issue to consider in the limitations section is that the data may not indicate brain differences that are specific to gymnasts. It is plausible that other world-class athletes in other sports may have similar brain differences compared to controls, and the study design does not permit this aspect to be discerned.

Major Point 2: There are several other individual characteristics to consider in matching controls, beyond age and sex. In particular, due to high level of physical fitness of world-class athletes, it is important to consider body-mass index (BMI) as a potential confound, because it can have a significant impact on white matter when imaged using diffusion MRI [1]. The other factor to consider is head size. Because fiber count was an outcome of the results, it is important to ensure that, for whatever reason, the gymnasts didn't have different intra-cranial volume. It is common practice to perform multiple linear regression with these two factors as covariates, and I would strongly recommend the authors consider expanding their statistical analysis to implement this approach, or something to the same effect.

Minor Issues:

* On line 24: it would be helpful to have a reference supporting the point about genetic predisposition. This could be an important consideration about causal nature of the effects.

* On line 31: extra left parenthesis.

* It would be good to include references to relevant manuscripts for the MRtrix tools.

* It may be good to cite the recent evaluation study by Schilling et al. [2] which discusses tractography limitations that might impact structural connectome analysis.

[1] Dekkers, I. A., Jansen, P. R., & Lamb, H. J. (2019). Obesity, brain volume, and white matter microstructure at MRI: a cross-sectional UK Biobank study. *Radiology*, 291(3), 763-771.

[2] Schilling, K. G., Nath, V., Hansen, C., Parvathaneni, P., Blaber, J., Gao, Y., ... & Landman, B. A. (2019). Limits to anatomical accuracy of diffusion tractography using modern approaches. *NeuroImage*, 185, 1-11.

Authors' Response

April 20, 2021

Cristina A. Ghiani, PhD, and J. Paula Warrington, PhD

Editors-in-Chief

Journal of Neuroscience Research

Dear Editors:

On behalf of all co-authors, I am pleased to submit the revised version of our manuscript (jnr-2021-Jan-9438) entitled “Connectome analysis of male world-class gymnasts using probabilistic multishell, multitissue constrained spherical deconvolution tracking,” which we request you to consider for publication

as an *Original Research* in *Journal of Neuroscience Research*.

We are immensely grateful for the time and effort taken by the editorial team and reviewer in improving our manuscript. We are also delighted that the reviewers found our work very interesting. The comments provided have been extremely useful in improving the quality of this manuscript. We believe that these revisions have substantially improved the quality of our work. The manuscript has been carefully rechecked

and appropriate changes have been made in accordance with the suggestions from the editors and reviewer.

Our point-by-point responses are provided below. We have highlighted (red bold font) all our revisions for

your convenience (as can be in the main document: marked revision). We hope that the revised manuscript

is now suitable for publication in your journal.

Notably, to be in line with the National Institutes of Health mandate to include sex as a biological variable,

we added “male” in the title. Following a reviewer’s recommendation, we added age, body mass index, and

intracranial volume as covariates in network-based statistic and graph theory analyses for possible confounding factors in brain microstructural changes. There were minimal changes in the results, leaving an unaltered discussion. All changes due to the re-analyses have been described in the revised manuscript. With agreement of all authors, we also changed the order of the first and second authors (upon revision:

1st

author, H.T., 2nd author, K.K.; both authors contributed equally). A letter of agreement signed by all authors

was submitted with the revised manuscript.

This manuscript has not been published elsewhere and is not under consideration by another journal. Each of the authors has approved the final version of the manuscript, agree with this submission to *Journal of Neuroscience Research*, and report no conflicts of interest. Moreover, this manuscript has been carefully reviewed by an experienced editor (www.enago.jp) who specializes in editing papers written by scientists whose native language is other than English.

This study was partially supported by the Private University Research Branding Project (Ministry of Education, Culture, Sports, Science and Technology, Japan) and JSPS KAKENHI (grant JP18K09005).

On behalf of the authors, I look forward to hearing from you at your earliest convenience.

Sincerely,

Koji Kamagata, MD, PhD

Department of Radiology

Juntendo University Graduate School of Medicine

2-1-1 Hongo, Bunkyo, Tokyo, 113-8421, Japan

Tel: +81-3-5802-1230

Fax: +81-3-3816-0958

Email: kkamagat@juntendo.ac.jp

Editorial Comments to the Author:

1. Please upload a graphical abstract, which we are asking of all authors submitting original research articles. This is intended to provide readers with a visual representation of the conclusions and an additional way to access the contents and appreciate the main message of the work. What we require is a .tif image file and a .doc text file containing an abbreviated abstract. For the image, labels, although

useful, must be kept to a minimum and the image should be 400 x 300, 300 x 400, or 400 x 400 pixels square and at a resolution of 72 dpi. This can be one of the figures from your article, or something slightly different, as long as it represents your study. Instructions for this can be found in our author guidelines online at [http://onlinelibrary.wiley.com/journal/10.1002/\(ISSN\)1097-4547/homepage/ForAuthors.html](http://onlinelibrary.wiley.com/journal/10.1002/(ISSN)1097-4547/homepage/ForAuthors.html)

Response:

A graphical abstract has been constructed according to the guidelines and uploaded with the revised manuscript.

Graphical abstract

Using the MSMT-CSD-based connectome, we identified increased structural connectivity in Japanese male world-class gymnasts (WCGs), involving sensorimotor, default-mode, attentional, visual, and limbic subnetworks correlated with D-scores of floor, parallel bars, horizontal bar, and years of training. Our findings indicated brain anatomical network plasticity in WCGs resulting from long-term intensive training.

2. The National Institutes of Health now mandates the inclusion of sex as a biological variable. To conform with this mandate, the Journal of Neuroscience Research has established new policy (please see our editorial: <http://onlinelibrary.wiley.com/doi/10.1002/jnr.23979/full>) requiring all authors to ensure proper consideration of sex as a biological variable. Please ensure that: 1) Any paper utilizing subjects of one sex state the sex of the sample in the title and abstract; 2) The number of samples/subjects of each sex used in the research must be clearly stated in the methods section; 3) The inability for any reason to study sex differences where they may exist should be discussed as a study limitation. We are also encouraging authors to report exploratory analyses of potential sex differences in studies not explicitly designed to address them.

Response:

We thank the editor for the clarification. All subjects included in the current study were male; thus, we revised the title and abstract as follows:

Title

(Page 1, Line 1-2)

Connectome analysis of male world-class gymnasts using probabilistic multishell, multitissue constrained spherical deconvolution tracking

Abstract

(Page 3, Line 6-7)

The connectome was mapped in 10 Japanese **male** WCGs and in 10 age-matched **male** controls.

3. Please add to your paper (after the Discussion and Acknowledgments, immediately before the References) a conflict of interest statement and a statement of authors' contributions. The statement must follow the CRediT Taxonomy. You can find examples of such statements in the author guidelines on-line at [http://onlinelibrary.wiley.com/journal/10.1002/\(ISSN\)1097-4547/homepage/ForAuthors.html](http://onlinelibrary.wiley.com/journal/10.1002/(ISSN)1097-4547/homepage/ForAuthors.html)

Response:

A conflict of interest statement and a statement of authors' contributions have been added to the revised manuscript.

Competing interests

(Page 22, Line 4-5)

The authors declare no competing interests.

Author contributions

(Page 22, Line 7-12)

H.T., K.K., C.A., H.W., H.S., M.H., S.A., and H.N. contributed to the conception and design of the study. H.T., K.K., C.A., W.U., M.F., Y.T., T.M., M.L., A.H., S.F., A.W., T.A., and S.M. contributed to the data collection, acquisition, and analysis of data. H.T., K.K., C.A., W.U., and H.W. contributed to drafting the manuscript and preparing the figures. All authors have reviewed and approved the contents of the manuscript.

4. Authors must submit, in the main text of the document, a 100-word-maximum statement about the

significance of their research paper written at a level that is understandable to the general public and to scientists outside their field of specialty. This statement will be distinct in purpose from the abstract, with the primary goal of broadly explaining the relevance and importance of this work and how this work contributes to different diseases.

Response:

A statement about the significance of the current study has been added to the revised manuscript.

Significance Statement

(Page 4, Line 2-10)

Recent evidence showed that long-term gymnastic training might induce brain neural plasticity, resulting in enhanced performance and higher skill levels in world-class gymnasts (WCGs). To extend the current understanding of neural mechanisms that distinguish WCGs from nonathletes, we evaluated brain neural network “connectome” in 10 WCGs and 10 nonathletes. Our results suggest the plasticity of brain network topology in WCGs resulting from long-term intensive training. Our findings also indicated the association between increased structural connectivity in some brain structures and specific gymnastic skills as indicated by D-score for each apparatus (i.e., floor exercise, pommel horse, vaulting horse, and parallel bars).

5. Please consider changing the orientation of Figure 3. The panels are too small to read the details. It may be best to display 3 brains per row instead of 6. Please also increase the font size used in each figure/panel.

Response:

We thank the reviewer for the suggestion. Accordingly, we revised Figure 3 as follows:

Statistics Editor: McArthur, David

Comments to the Author:

MRtrix software is cited as available at brain.org.au but that website indicates that this version [0.2] "is no longer under active development and now only receives bug fix updates. The new MRtrix3 release, which includes the latest developments, is available on the MRtrix3 Website, along with its documentation." Presumably you are referring to version 0.2.

Response:

We thank the reviewer for identifying that MRtrix software is incorrectly defined and cited throughout the manuscript. Indeed, in the current study, we applied MRtrix3; thus, we revised the sentences as follows:

2.3. Preprocessing for connectome

(Page 9, Line 6-8)

The “5tt2gmwmi” command in the MRtrix3 software package (<https://www.mrtrix.org>) (Tournier et al., 2019) was utilized to acquire the GM–WM interface mask.

2.5 Defining edges

(Page 9, Line 12-Page 10, Line 2)

MSMT-CSD probabilistic tracking with multishell DW-MRI data (b-values of 0, 1000, and 2000 s/mm²) utilizing the MRtrix3 software package (Brain Research Institute, Melbourne, Australia, <http://www.brain.org.au/software/>) was used to acquire whole-brain tractograms.

(Page 10, Line 10-12)

Lastly, the *dwi2fod* command with the *msmt-csd* option in MRtrix3 was utilized to acquire fiber orientation distribution functions (fODF) of WM, GM, and CSF.

Table 4 shows a footnote "* Statistical significance" that appears to be unused and would not seem to be needed. The final p-value in that table is displayed to 4 decimal digits, which should read ">0.001" in keeping with all the other values (i.e., a stable degree of decimal exactitude).

Response:

According to the editor’s suggestion, we revised Table 4 (Page 34-35) as follows:

Table 4. Regions with a significant between-group difference in nodal strength and nodal degree

Regions Controls WCG *t* *P*-value** Cohen’s

d

Nodal strength

Right lateral orbitofrontal gyrus 3113.6

(251.28)

3882.2

(429.13)

4.64 0.002 2.19

Right temporal pole 866.9

(115.62)

1165.6

(187.53)

4.07 0.007 1.91

Right rostral middle frontal gyrus 7007.6

(883.57)

8596.1

(873.01)

3.84 0.001 1.81

Right pars orbitalis 1892.5

(229.18)

2390.8

(320.96)

3.79 0.001 1.79

Left temporal pole 1026.2

(257.68)

1437.1

(217.08)

3.66 0.002 1.72

Right precentral gyrus 10899.3

(919.52)

12479.5

(1014.8)

3.46 0.003 1.63

Right inferior temporal gyrus 5175.1

(758.2)

6248.9

(611.2)

3.31 0.004 1.56

Right inferior parietal gyrus 8250.1

(590.8)

9561.2

(1074.4)

3.21 0.005 1.51

Right superior frontal gyrus 13275

(938.3)

14833.8

(1128.5)

3.19 0.005 1.50

Right rostral anterior cingulate gyrus 1646.7

(207.4)

1969.9

(223.1)

3.18 0.005 1.50
 Right medial orbitofrontal gyrus 2209.1
 (253.9)
 2566.9
 (232.4)
 3.12 0.006 1.47
 Left precentral gyrus 11589.9
 (1151.3)
 13222.2
 (1123.2)
 3.04 0.006 1.44
 Nodal degree
 Left inferior parietal gyrus 55.7
 (3.38)
 62.2
 (2.4)
 4.71 0.002 2.22
 Right inferior parietal gyrus 58.2
 (2.44)
 63.1
 (2.39)
 4.31 0.0004
>0.001
 2.03

Notes: Data are expressed as mean (SD). * denotes statistical significance. **False discovery ratecorrected

P-values. *Abbreviations*: SD, standard deviation; WCG, world-class gymnasts.

Table S4 includes several *t*-value entries with a single decimal digit; these should be expressed as 2 digits in keeping with the other such values. The phrase "Identified connection ($P = 0.01$)" should match the narrative's value in Section 4.1 of " $P = 0.011$ ".

Response:

We revised Table S4 as follows:

Table S4. Network identified as significantly different between world-class gymnasts and controls using network-based statistical analysis

Identified connection ($P = 0.00120$) *t*-value

Left inferior parietal to right middle temporal **5.49**
 Right inferior temporal to right paracentral **5.46**
 Left superior parietal to right hippocampus **4.03**
 Right lingual to right precuneus **3.97**
 Left inferior parietal to right hippocampus **3.92**
 Left isthmus cingulate to right inferior temporal **3.88**
 Left thalamus proper to right inferior temporal **3.86**
 Right lateral occipital to right lingual **3.71**
 Left paracentral to left superior temporal **3.67**
 Left inferior parietal to right putamen **3.56**
 Right pars opercularis to right precentral **3.50**
 Left superior frontal to right pars orbitalis **3.49**
 Left insula to left accumbens area **3.47**
 Left bankssts to right posterior cingulate **3.47**
 Right pars orbitalis to right precentral **3.45**

Right lateral orbitofrontal to right paracentral **3.42**
Right inferior temporal to right isthmus cingulate **3.40**
Left middle temporal to right paracentral **3.38**
Right medial orbitofrontal to right pars orbitalis **3.38**
Right lateral orbitofrontal to right pars orbitalis **3.36**
Left pars triangularis to left transverse temporal **3.35**
Left lateral orbitofrontal to right rostral middle frontal **3.34**
Left paracentral to left superior parietal **3.32**
Left inferior parietal to right inferior temporal **3.27**
Right caudal middle frontal to right pars triangularis **3.27**
Left superior parietal to right inferior temporal **3.25**
Left precuneus to right inferior temporal **3.23**
Left superior temporal to right hippocampus **3.20**
Right inferior parietal to right inferior temporal **3.15**
Left putamen to right medial orbitofrontal **3.14**
Right superior temporal to right transverse temporal **3.14**
Left bankssts to left putamen **3.12**
Right inferior parietal to right pars orbitalis **3.06**
Right inferior temporal to right superior parietal **3.04**
Left middle temporal to left insula **3.03**
Right inferior temporal to right precuneus **3.02**
Right inferior temporal to right middle temporal **2.99**
Right pars opercularis to right pars orbitalis **2.98**
Right putamen to right lateral orbitofrontal **2.96**
Left inferior parietal to right lateral occipital **2.95**
Left middle temporal to left pars triangularis **2.94**
Right caudate to right lateral orbitofrontal **2.93**
Left pars triangularis to left superior temporal **2.92**
Right lateral orbitofrontal to right superior parietal **2.91**
left rostral caudal middle frontal to left thalamus proper **2.84**
Left insula to right rostral anterior cingulate **2.82**
Left medial orbitofrontal to right rostral middle frontal **2.82**
Right inferior temporal to right transverse temporal **2.82**
Right bankssts to right middle temporal **2.81**
Left inferior parietal to right pars opercularis **2.81**
Right caudal middle frontal to right pars opercularis **2.81**
Left lateral orbitofrontal to right medial orbitofrontal **2.80**
Left superior frontal to right caudal middle frontal **2.76**
Left thalamus proper to right fusiform **2.72**
Right pars triangularis to right precentral **2.71**
Right lateral orbitofrontal to left frontal pole **2.68**
Left superior frontal to right superior frontal **2.68**
Left inferior parietal to right lingual **2.66**
Left bankssts to left posterior cingulate **2.65**
Right precentral to right superior frontal **2.65**
Left amygdala to right posterior cingulate **2.64**
Right lateral orbitofrontal to right rostral middle frontal **2.63**
Right caudal anterior cingulate to right rostral anterior cingulate **2.62**
Left inferior temporal to right hippocampus **2.61**
Left lateral occipital to left amygdala **2.60**

Right entorhinal to right middle temporal **2.60**

Right pars orbitalis to right postcentral **2.59**

Reviewer: 1

Comments to the Author

This manuscript presents work investigating structural brain differences between world-class gymnasts and matched controls, as assessed by magnetic resonance imaging. In particular, the study focuses on network-level effects derived from tractography-based modeling of brain connectivity between cortical brain areas. Overall, I found the paper to be interesting and well-written, and the technical aspects of the image analysis reflect the state-of-the-art.

I have several concerns to note:

Major Point 1: The manuscript makes several statements regarding neural plasticity and long-term intensive training, but I think the paper should more carefully note the limitations of the present study design. Specifically, because of the cross-sectional nature of the study, the data cannot tell us whether the brain differences are due to training, genetics, early life environment, etc. The discussion points out this limitation, but the abstract may lead the reader to think otherwise, so I think it would improve the manuscript to soften some of these claims. I think one other issue to consider in the limitations section is that the data may not indicate brain differences that are specific to gymnasts. It is plausible that other world-class athletes in other sports may have similar brain differences compared to controls, and the study design does not permit this aspect to be discerned.

Response:

We thank the reviewer for the insightful suggestions. Accordingly, we revised the manuscript as follows:

Abstract

(Page 3, Line 16-22)

The brain topology changes demonstrated in the current study may represent the neural basis for outstanding gymnastics performance resulting from brain plasticity. **Together, these findings extend the current understanding of neural mechanisms that distinguish WCGs from controls and suggest brain anatomical network plasticity in WCGs resulting from long-term intensive training. Future studies should assess the contribution of genetic or early-life environmental factors in the brain network organization of WCGs.**

5.5 Limitations

(Page 20, Line 26-Page 21, Line 3)

In addition, an additional longitudinal study with a larger sample size is needed to determine whether genetic **or early-life environmental factors** influence the brain network topology of WCGs and/or whether long-term intensive training is a major contributor to the changes.

(Page 21, Line 4-7)

It is also worth noting that brain anatomical network plasticity shown in this study might not be specific to WCGs. Further studies should also evaluate brain network organization in world-class athletes with different sports.

Major Point 2: There are several other individual characteristics to consider in matching controls, beyond age and sex. In particular, due to high level of physical fitness of world-class athletes, it is important to consider body-mass index (BMI) as a potential confound, because it can have a significant impact on white matter when imaged using diffusion MRI [1]. The other factor to consider is head size. Because fiber count was an outcome of the results, it is important to ensure that, for whatever reason, the gymnasts didn't have different intra-cranial volume. It is common practice to perform

multiple linear regression with these two factors as covariates, and I would strongly recommend the authors consider expanding their statistical analysis to implement this approach, or something to the same effect.

Response:

According to the reviewer's suggestion, we included age, BMI, and ICV as covariates in the statistical analyses of NBS and graph theory. There were minimal changes in the results of NBS and global topographic metrics, leaving an unaltered discussion. However, between-group comparison of local topographical metrics lost statistical significance status after age, BMI, and ICV were added as covariates. We suspect that the small sample size led to decreased statistical power. Nevertheless, the outcome with age as a covariate was the same as the original results. The changes in the revised manuscript are described below.

2.2. Image acquisition

(Page 8, Line 14-16)

To enable the estimation of **intracranial volume (ICV)** and cortical parcels and tissue segmentation by FreeSurfer, T1-weighted images (T1WI) were also acquired using three-dimensional (3D) magnetizationprepared rapid gradient-echo sequence.

2.8 Detection of disrupted WM connectivity

(Page 12, Line 3-7)

As a summation, a two-sample *t* test **An analysis of covariance (ANCOVA)** was performed independently at each edge to test the null hypothesis of equivalence of the mean streamline count of WCGs and controls. **Age, body mass index (BMI), and ICV were included as nuisance covariates for possible confounding factors in brain microstructural changes (Dekkers, Jansen, & Lamb, 2019; Kijonka et al., 2020).**

3. Statistical analysis

(Page 12, Line 20-24)

In accordance with the Kolmogorov–Smirnov test, between-group differences were analyzed by Student *t*-tests for age, **BMI, and ICV** global, and local topological metrics and chi-squared tests for gender.

Further, the differences in global and local topological metrics were assessed using ANCOVA with age, BMI, and ICV as nuisance covariates.

4.1. Changes in WM connection characteristics of Japanese WCGs

(Page 13, Line 10-21)

Specifically, the NBS identified significantly increased subnetwork connectivity comprising 64 **67** edges and 49 **53** nodes in the WCG group relative to the control group ($P = 0.011$ **0.020**). We did not detect significantly decreased subnetwork connectivity in WCGs relative to controls.

All edges with increased connectivity in WCGs were connected to brain areas that were classified as sensorimotor, attention, visual, limbic/subcortical, and default-mode systems. These 10 **19** edges can be classified into intrasystem connections and 54 **48** intersystem connections. Among the intrasystem connections, four **eight** edges were connected within attention areas, three edges were connected within the sensorimotor areas, and two **seven** edges were connected within default-mode areas, **and one edge was connected with visual area.** For the intersystem connections, 49 **33** of 54 **48** edges were connected between the sensorimotor, attention, and default-mode, **and visual** systems.

4.4. Local metric characteristics of WCGs

(Page 15, Line 7-11)

No between-group differences in local topological metrics were observed after adjusting for age, BMI, and ICV. However, with only age included as a covariate, 12 brain regions with significantly higher nodal strength and two brain regions with significantly higher nodal degree were detected in the WCGs compared with the controls (Figure 4; Table 4).

5. Discussion

5.1. Changes in the WM connection characteristics of the WCGs

(Page 16, Line 10-15)

We found that all 64 **67** edges with increased connectivity in WCGs were connected to brain regions that were classified to the sensorimotor, attention, visual, limbic/subcortical, and default-mode systems. Specifically, the edges consisted of 10 **19** intrasystem connections and 54 **48** intersystem connections. Interestingly, all 10**19** intrasystem connections are within the sensorimotor, attention, **visual**, and defaultmode system, and most of the 54 intersystem connections (49/54 **33/48**) link the sensorimotor, attention, **visual**, and default-mode systems.

5.3. Global and local metric characteristics of the WCGs

(Page 19, Line 8-10)

At the local level, **after adjusting for age**, WCGs show increased “functional importance,” as indicated by increased nodal strength in the sensorimotor, default-mode, attention, and limbic/subcortical system compared with controls.

Figure 2.

Page 29-31

Table 1. Demographic characteristics of world-class gymnasts

World- D-Score

class

gymnasts

Best medal record

Age

(years)

Years of

training

BMI ICV (mL) Floor

exercise

Pommel

horse

Rings Vault

Parallel

bars

Horizontal

bar

Mean ±

SD

1

DTB team challenge

2017 bronze medal.

20 14 **20.6 1457.6** 5.3 6.0 5.6 5.2 5.8 5.6

5.58 ±

0.30

2

WC2018 Tokyo cup

silver medal

(WGC2018 bronze

medal)

21 14 **21.0 1459.8** 6.0 5.8 6.0 5.6 6.2 5.1

5.78 ±

0.39

3

WGC2015 gold medal

(WGC2018 bronze

medal)

21 12 **22.5 1623.3** 5.9 6.4 6.1 5.2 6.3 5.7

5.93 ±

0.44

4 Asian Games 2018

silver medal

Universiade2017 gold

21 19 **22.5 1520.3** 6.0 6.0 5.7 5.2 6.2 5.7 5.80 ±

0.35

medal

5 WGC2015 gold medal 22 13 **22.04 1582.4** 6.2 5.4 5.3 5.2 5.4 5.9

5.57 ±

0.39

6

DTB team challenge

2018 bronze medal

20 10 **20.8 1502.7** 5.9 6.3 5.6 5.2 5.7 5.2

5.65 ±

0.42

7

DTB team challenge

2018 bronze medal

19 14 **24.0 1499.2** 5.5 5.1 5.6 5.2 5.9 6.1

5.57 ±

0.39

8

Voronin cup 2016

gold medal

19 14 **23.4 1349.7** 5.6 5.8 5.5 4.8 6.0 5.8

5.58 ±

0.42

9 IJGC 2015 gold medal 18 14 **22.3 1482.2** 5.9 5.7 5.0 5.6 5.5 4.8

5.42 ±

0.43

10

Asian Games 2018

silver medal

(All Japan2018 gold

medal)

18 12 **21.8 1588.0** 5.9 6.1 5.3 5.2 6.0 5.6

5.68 ±

0.38

Mean ± SD 19.9 ±

1.4

13.6 ±

2.3

22.9

± 1.0

1506.5 ±

74.9

5.82 ±

0.27

5.86 ±

0.39

5.57 ±

0.33

5.24

±

5.90 ±

0.30

5.55 ± 0.40

Abbreviations: **BMI, body mass index**; D-score, difficulty score; DTB, Deutscher Turner-Bund; **ICV, intracranial volume**; IJGC, International

Junior Gymnastics Competition; SD, standard deviation; WC, World Cup; WGC, World Gymnastics Championships.

All WCGs in this study have won medals in gymnastics world championships since 2015.

0.23

Page 33

Table 3. Between-group comparison of global network measures

Controls World-class

gymnasts

F *P*-value* Partial η^2

Global clustering

(SD)

0.0051

(0.0007)

0.0049

(0.0008)

0.034 0.86 0.0023

Global efficiency

(SD)

0.040

(0.006)

0.041

(0.007)

0.86 0.43 0.054

Global strength

(SD)

4751.48

(295.92)

5205.66

(329.46)

18.40 0.0050 0.55

Characteristic path

length

(SD)

0.0064

(0.0004)
0.0057
(0.0005)
13.44 0.0080 0.47
 Small-worldness ratio
 (SD)
2.34
(0.44)
2.62
(0.56)
2.82 0.16 0.16

Note. Data are expressed as mean (SD). *False discovery rate-corrected P-values.
 Abbreviation: SD, standard deviation.

Supporting Information Tables

Table S2. Cohen's *d* of each global metric across the full range of sparsity thresholds for comparison between the world-class gymnasts and controls

Partial eta-squared

Threshold	100%	5%	10%	15%	20%	25%	30%
Mean strength	0.55	0.56	0.54	0.54	0.54	0.54	0.54
Global clustering	0.0020	0.0050	0.039	0.045	0.045	0.049	0.038
Global efficiency	0.054	0.061	0.055	0.054	0.054	0.054	0.054
Characteristic path length	0.47	0.46	0.44	0.47	0.47	0.47	0.47
Small-world property	0.16	0.18	0.22	0.21	0.20	0.20	0.20

Table S3. Networks identified as significantly different between world-class gymnasts and controls using network-based statistical analysis

$P = 0.05$

T = 1.75

Network 1

$P = 0.02$

T = 2.24

Network 1

$P = 0.01$

T = 2.58

Network 1

$P = 0.005$

T = 2.92

Table S4. Network identified as significantly different between world-class gymnasts and controls using network-based statistical analysis

Identified connection ($P = 0.001$) *t*-value

Left inferior parietal to right middle temporal **5.49**

Right inferior temporal to right paracentral **5.46**

Left superior parietal to right hippocampus **4.03**

Right lingual to right precuneus **3.97**

Left inferior parietal to right hippocampus **3.92**

Left isthmus cingulate to right inferior temporal **3.88**

Left thalamus proper to right inferior temporal **3.86**

Right lateral occipital to right lingual **3.71**

Left paracentral to left superior temporal **3.67**

Left inferior parietal to right putamen **3.56**

Right pars opercularis to right precentral **3.50**
Left superior frontal to right pars orbitalis **3.49**
Left insula to left accumbens area **3.47**
Left bankssts to right posterior cingulate **3.47**
Right pars orbitalis to right precentral **3.45**
Right lateral orbitofrontal to right paracentral **3.42**
Right inferior temporal to right isthmus cingulate **3.40**
Left middle temporal to right paracentral **3.38**
Right medial orbitofrontal to right pars orbitalis **3.38**
Right lateral orbitofrontal to right pars orbitalis **3.36**
Left pars triangularis to left transverse temporal **3.35**
Left lateral orbitofrontal to right rostral middle frontal **3.34**
Left paracentral to left superior parietal **3.32**
Left inferior parietal to right inferior temporal **3.27**
Right caudal middle frontal to right pars triangularis **3.27**
Left superior parietal to right inferior temporal **3.25**
Left precuneus to right inferior temporal **3.23**
Left superior temporal to right hippocampus **3.20**
Right inferior parietal to right inferior temporal **3.15**
Left putamen to right medial orbitofrontal **3.14**
Right superior temporal to right transverse temporal **3.14**
Left bankssts to left putamen **3.12**
Right inferior parietal to right pars orbitalis **3.06**
Right inferior temporal to right superior parietal **3.04**
Left middle temporal to left insula **3.03**
Right inferior temporal to right precuneus **3.02**
Right inferior temporal to right middle temporal **2.99**
Right pars opercularis to right pars orbitalis **2.98**
Right putamen to right lateral orbitofrontal **2.96**
Left inferior parietal to right lateral occipital **2.95**
Left middle temporal to left pars triangularis **2.94**
Right caudate to right lateral orbitofrontal **2.93**
Left pars triangularis to left superior temporal **2.92**
Right lateral orbitofrontal to right superior parietal **2.91**
left rostral caudal middle frontal to left thalamus proper **2.84**
Left insula to right rostral anterior cingulate **2.82**
Left medial orbitofrontal to right rostral middle frontal **2.82**
Right inferior temporal to right transverse temporal **2.82**
Right bankssts to right middle temporal **2.81**
Left inferior parietal to right pars opercularis **2.81**
Right caudal middle frontal to right pars opercularis **2.81**
Left lateral orbitofrontal to right medial orbitofrontal **2.80**
Left superior frontal to right caudal middle frontal **2.76**
Left thalamus proper to right fusiform **2.72**
Right pars triangularis to right precentral **2.71**
Right lateral orbitofrontal to left frontal pole **2.68**
Left superior frontal to right superior frontal **2.68**
Left inferior parietal to right lingual **2.66**
Left bankssts to left posterior cingulate **2.65**
Right precentral to right superior frontal **2.65**

Left amygdala to right posterior cingulate **2.64**

Right lateral orbitofrontal to right rostral middle frontal **2.63**

Right caudal anterior cingulate to right rostral anterior cingulate **2.62**

Left inferior temporal to right hippocampus **2.61**

Left lateral occipital to left amygdala **2.60**

Right entorhinal to right middle temporal **2.60**

Right pars orbitalis to right postcentral **2.59**

Minor Issues:

* On line 24: it would be helpful to have a reference supporting the point about genetic predisposition. This could be an important consideration about causal nature of the effects.

Response:

We thank the reviewer for the suggestion. Accordingly, we cited some relevant studies in the revised manuscript.

1. Introduction

(Page 5, Line 8-11)

Genetic predispositions can make individual differences in athletic performance learning (**Ahmetov, Egorova, Gabdrakhmanova, & Fedotovskaya, 2016; Guth & Roth, 2013; Yan, Papadimitriou, Lidor, & Eynon, 2016**), but there is no doubt that achieving incredible athletic performance requires long and intensive training.

* On line 31: extra left parenthesis.

Response:

Thank you for pointing out the error. In the revised manuscript, we have deleted the extra left parenthesis.

2.2. Image acquisition

(Page 8, Line 1-2)

Structural MRI and DW-MRI were performed using a 3T MR scanner (MAGNETOM Prisma; Siemens Healthcare, Erlangen, Germany) with a 64-channel head coil.

* It would be good to include references to relevant manuscripts for the MRtrix tools.

Response:

As described in <https://mrtrix.readthedocs.io/en/latest/>, we cited a manuscript from “J.-D. Tournier, R. E. Smith, D. Raffelt, R. Tabbara, T. Dhollander, M. Pietsch, D. Christiaens, B. Jeurissen, C.-H. Yeh, and A. Connelly. *MRtrix3: A fast, flexible and open software framework for medical image processing and visualisation*. *NeuroImage*, 202 (2019), pp. 116–37.” in the revised manuscript.

2.3. Preprocessing for connectome

(Page 9, Line 6-8)

The “5tt2gmwmi” command in the MRtrix3 software package (<https://www.mrtrix.org>) (**Tournier et al., 2019**) was utilized to acquire the GM–WM interface mask.

* It may be good to cite the recent evaluation study by Schilling et al. [2] which discusses tractography limitations that might impact structural connectome analysis.

Response:

Accordingly, we cited the study by Schilling et al. in the revised manuscript as follows:

1. Introduction

(Page 6, Line 14-16)

Furthermore, tractography is known to be heavily affected by the quality of the diffusion MRI acquisition leading to false-positive or false-negative connections (Schilling et al., 2019).

[1] Dekkers, I. A., Jansen, P. R., & Lamb, H. J. (2019). Obesity, brain volume, and white matter microstructure at MRI: a cross-sectional UK Biobank study. *Radiology*, 291(3), 763-771.

[2] Schilling, K. G., Nath, V., Hansen, C., Parvathaneni, P., Blaber, J., Gao, Y., ... & Landman, B. A. (2019). Limits to anatomical accuracy of diffusion tractography using modern approaches. *NeuroImage*, 185, 1-11.

2nd Editorial Decision

Dear Dr Kamagata:

Thank you for submitting your manuscript "Connectome analysis of male world-class gymnasts using probabilistic multishell, multitissue constrained spherical deconvolution tracking" by Tomita, Hiroyuki; Kamagata, Koji; Andica, Christina; Uchida, Wataru ; Fukuo, Makoto; Waki, Hidefumi; Sugano, Hidenori ; Tange, Yuichi; Mitsuhashi, Takumi; Lukies, Matthew; Hagiwara, Akifumi; Fujita, Shohei; Wada, Akihiko; Akashi, Toshiaki ; Murata, Syo; Harada, Mutsumi; Aoki, Shigeki; Naito, Hisashi.

You will be pleased to know that your manuscript has been accepted for publication. Thank you for submitting this excellent work to our journal.

In the coming weeks, the Production Department will contact you regarding a copyright transfer agreement and they will then send an electronic proof file of your article to you for your review and approval.

Please note that your article cannot be published until the publisher has received the appropriate signed license agreement. Within the next few days, the corresponding author will receive an email from Wiley's Author Services asking them to log in. There, they will be presented with the appropriate license for completion. Additional information can be found at <https://authorservices.wiley.com/author-resources/Journal-Authors/licensing-open-access/index.html>

Would you be interested in publishing your proven experimental method as a detailed step-by-step protocol? *Current Protocols in Neuroscience* welcomes proposals from prospective authors to disseminate their experimental methodology in the rapidly evolving field of neuroscience. Please submit your proposal here: <https://currentprotocols.onlinelibrary.wiley.com/hub/submitproposal>

Congratulations on your results, and thank you for choosing the *Journal of Neuroscience Research* for publishing your work. I hope you will consider us for the publication of your future manuscripts.

Sincerely,

Dr Sandra Chanraud
Associate Editor, *Journal of Neuroscience Research*

Dr Junie Warrington
Editor-in-Chief, *Journal of Neuroscience Research*

Associate Editor: Chanraud, Sandra
Comments to the Author:
(There are no comments.)

Reviewer: 1

Comments to the Author

The authors have addressed my concerns with the revised manuscript, and I thank them for their additions to expand their analysis and clarify the points raised.

ors' Response Auth

3rd Editorial Decision

Authors' Response

2nd Editorial Decision

Decision Letter

Authors' Response

3rd Editorial Decision

Decision Letter

Authors' Response

4th editorial decision

Decision Letter

Author response